# Continual Learning through Control Minimization

**Sander de Haan** [1 2]   **Yassine Taoudi-Benchekroun** [1 2]   **Pau Vilimelis Aceituno** [1 2 †]   **Benjamin F. Grewe** [1 2 †]

## Abstract

Catastrophic forgetting remains a fundamental challenge for neural networks when tasks are trained sequentially. In this work, we reformulate continual learning as a control problem where learning and preservation signals compete within neural activity dynamics. We convert regularization penalties into preservation signals that protect prior-task representations. Learning then proceeds by minimizing the control effort required to integrate new tasks while competing with the preservation of prior tasks. At equilibrium, the neural activities produce weight updates that implicitly encode the full prior-task curvature, a property we term the *continual-natural gradient*, requiring no explicit curvature storage. Experiments confirm that our learning framework recovers true prior-task curvature and enables task discrimination, outperforming existing methods on standard benchmarks without replay.

## 1. Introduction

Catastrophic forgetting remains a fundamental challenge for neural networks when tasks are trained sequentially (McCloskey & Cohen, 1989; French, 1999). One approach to mitigating forgetting is to identify which parameters matter for previously learned tasks and penalize changes to those parameters during subsequent learning (van de Ven & Tolias, 2019; de Lange et al., 2019). Parameter importance corresponds to the curvature of the loss landscape with respect to each parameter, measuring how sensitive the previous task's loss is to changes in that parameter. While the Hessian characterizes the full curvature, computing and storing the complete Hessian is intractable for modern networks. Practical methods therefore approximate the curvature through the Fisher information matrix (Benzing, 2022), which equals the Hessian at a strict local minimum for standard supervised losses. Research on regularization methods has focused on obtaining better curvature approximations. Early methods store the diagonal of the Fisher (Kirkpatrick et al., 2017; Liu et al., 2018), capturing per-parameter sensitivity but discarding interactions between parameters. Kronecker-factored approaches approximate the Fisher as a block-diagonal matrix, retaining within-layer parameter interactions through structured factorizations of the Fisher (Martens & Grosse, 2015; Ritter et al., 2018). Online methods accumulate parameter importance during training rather than computing curvature at fixed points (Zenke et al., 2017; Aljundi et al., 2018; Schwarz et al., 2018), and quasi-Newton methods maintain low-rank curvature estimates that evolve as learning progresses (Vander Eeckt & Van Hamme, 2025). Gradient-based methods take a complementary approach by constraining parameter updates to directions that do not interfere with previously learned tasks, requiring explicit storage of gradient subspaces computed at task boundaries (Saha et al., 2021).

All parameter-based regularization methods share two fundamental limitations. Curvature estimates remain fixed or evolve only with respect to the current task, while sequential training moves parameters through regions where the geometry of previous tasks changes, thereby increasing misalignment between the estimated and true prior-task curvature (Wu et al., 2024). Regularization methods also fail in settings requiring task discrimination (Kim et al., 2022), even when curvature approximations are improved (Masana et al., 2023). We observe that both limitations arise because the regularization methods described above operate separately from the learning process and apply corrections after gradients have been computed, leaving the learning of new tasks decoupled from the preservation of prior tasks. If learning and preservation signals could interact during the formation of learning signals, new-task learning and prior-task preservation would no longer be decoupled. A separate line of bio-inspired research offers precisely the machinery that we require by embedding error information directly into neural activity dynamics rather than propagating errors backward (Rao & Ballard, 1999; Guerguiev et al., 2017; Whittington & Bogacz, 2017; Scellier & Bengio, 2017; Song et al., 2024; Aceituno et al., 2025).

---

[†]Equal supervision  [1]Institute of Neuroinformatics, University of Zurich and ETH Zurich, Zurich, Switzerland [2]ETH AI Center, ETH Zurich, Zurich, Switzerland. Correspondence to: Sander de Haan <sdehaan@ethz.ch>, Benjamin F. Grewe <bgrewe@ethz.ch>.

*Proceedings of the $43^{rd}$ International Conference on Machine Learning*, Seoul, South Korea. PMLR 306, 2026. Copyright 2026 by the author(s).

Control-based formulations instantiate this approach by introducing neuron-specific control signals that modulate neural activities, driving the network toward states that reduce the loss (Meulemans et al., 2021; 2022a). Meulemans et al. (2022b) introduce the least-control principle, which rethinks learning as minimizing the control effort required to reach a loss-minimizing equilibrium rather than minimizing the loss directly. The least-control objective acts as a budget that the control signal must allocate selectively, and the network dynamics determine which neurons are inexpensive to modulate. Once the network dynamics reach equilibrium, parameters update to make desired equilibria easier to reach, reducing the control effort over time. Learning therefore proceeds within the geometry imposed by the network dynamics.

In this work, we reformulate continual learning as a control minimization problem by converting parameter-space regularization penalties into activity-space preservation signals. Intuitively, preservation increases the cost of modulating neurons in ways that would interfere with previously learned representations, naturally steering the control signal away from those directions. Our framework, termed Equilibrium Fisher Control (EFC), couples new-task learning to prior-task preservation by making learning and preservation signals compete within neural dynamics. At equilibrium, the neural activities produce weight updates that implicitly encode the full prior-task curvature without the need for explicit storage. We term this the *continual-natural gradient*, as learning now proceeds within the geometry of previously learned tasks. Experiments confirm that our learning framework recovers true prior-task curvature and enables task discrimination, outperforming existing parameter-based regularization methods on standard benchmarks without replay.

## 2. The Equilibrium Fisher Control framework

Here, we introduce Equilibrium Fisher Control (EFC), a framework that reformulates continual learning as competition between learning and preservation signals within neural activity dynamics. The central idea is to convert parameter-space regularizers into activity-dependent preservation signals that modulate neural activities, and forcing the learning signal to compete with preservation before parameters update. We introduce the preservation signal in Section 2.1, the resulting network dynamics in Section 2.2, and the control minimization learning objective in Section 2.3. Full derivations appear in Supplementary A.

### 2.1. Preservation signal

We can transform any parameter-based regularizer $\mathcal{R}(\theta)$ into a neuron-specific preservation signal $\gamma$ by projecting the regularizer gradient onto current presynaptic activity. For neuron $k$ with presynaptic activity $\phi_k$ and incoming synaptic parameters $\theta_k$,

$$\gamma_k = \phi_k^\top \nabla_{\theta_k} \mathcal{R}(\theta) \tag{1}$$

which we instantiate using the canonical EWC regularizer (Kirkpatrick et al. (2017), Supplementary A.1),

$$\gamma_k = \beta \, \phi_k^\top F_{A,k}^D \left( \theta_k - \theta_{A,k}^* \right) \tag{2}$$

where $\beta > 0$ controls preservation strength and $F_{A,k}^D$ is the diagonal Fisher information matrix restricted to neuron $k$'s incoming parameters. This preservation signal activates only when the current input engages synapses that were relevant to previously learned tasks. Most parameter-based regularization methods approximate the Fisher information in some form (Benzing, 2022), and our construction generalizes accordingly.

### 2.2. Network dynamics

We now embed this preservation signal within a dynamical neural network where learning and preservation compete within neural activities. The network state $\phi \in \mathbb{R}^N$ concatenates neural activities across all layers and evolves according to,

$$\tau \dot{\phi} = -\phi + e^{\psi + \gamma} \odot f(\phi, \theta) \tag{3}$$

where $f(\phi, \theta)$ denotes the feedforward mapping, $\psi$ is a neuron-specific learning signal that drives the system toward low loss, $\gamma$ is the preservation signal defined above, and $\odot$ represents element-wise multiplication. The exponential multiplicative form ensures that modulation scales with the current neural activity and that learning and preservation are coupled rather than independent from each other. Therefore, a strongly active neuron receives proportionally stronger modulation from both signals. In the absence of both signals ($\psi = \gamma = 0$) or when preservation and learning oppose each other ($\psi = -\gamma$), the equilibrium $\phi_\star = f(\phi_\star, \theta)$ recovers standard feedforward computation. Because both signals modulate the feedforward activity $f(\phi, \theta)$, learning can only overcome preservation by expending opposing effort. Standard parameter-based regularization lacks this property, applying identical corrections regardless of neural activity.

### 2.3. Learning objective

We seek the minimal learning signal required to reach a loss-minimizing equilibrium while competing with the preservation signal. To formalize this objective, we extend the least-control principle of Meulemans et al. (2022b) to a multiplicative formulation (see Supplementary A.2),

$$\min_\psi \|\psi\|^2 \text{ s.t. } \underbrace{\phi = e^{\psi + \gamma} \odot f(\phi, \theta)}_{\text{equilibrium of } \phi}, \underbrace{\nabla_\phi \mathcal{L}(\phi) = \mathbf{0}}_{\text{loss minimum}} \tag{4}$$

where the minimal-norm objective forces the learning signal to allocate effort selectively. The preservation signal $\gamma$ raises

the cost of modulating specific neurons, and $\psi_\star$ must find the lowest-cost path to a loss-minimizing state. Directions that would modify preserved representations become expensive, while directions orthogonal to previous learning remain cheap.

The learning procedure has two stages. First, with parameters $\theta$ fixed, run the controlled dynamics until convergence to obtain the optimal learning signal $\psi_\star$ and equilibrium activities $\phi_\star$. Second, update $\theta$ by descending $\mathcal{H}(\theta) \triangleq \|\psi_\star(\theta)\|^2$ (see Supplementary A.3 for gradients). Intuitively, parameters gradually relax toward values requiring ever-smaller learning effort to reach equilibrium (Rao & Ballard, 1999; Meulemans et al., 2022a; Song et al., 2024; Aceituno et al., 2025). We instantiate this objective using a feedback controller that drives the network toward equilibrium[1]. Supplementary A.4 describes the controller dynamics, Supplementary A.5 derives the general controller class, and Supplementary A.6 compares with the original additive least-control formulation of Meulemans et al. (2022b).

# 3. Learning theory

Here, we formalize the learning-theoretic properties of EFC. We establish that learning at equilibrium allows the network dynamics to implicitly encode an approximation of the full prior-task curvature in Section 3.1, a property we term the continual-natural gradient. We show that this curvature-aware learning enables task discrimination by filtering interference from gradients that regularization methods cannot remove in Section 3.2. We analyze forgetting bounds and demonstrate that EFC outperforms regularization-based approaches even when regularizers have access to the full prior-task curvature in Section 3.3. Detailed derivations and proofs appear in Supplementary B, C, D, and E.

## 3.1. The continual-natural gradient property

Our learning framework updates weights at the equilibrium point $(\psi_\star, \phi_\star)$, reached by running the dynamics until convergence (see Supplementary B for steady-state analysis). We will show that Learning at equilibrium captures second-order parameter interactions without storing or computing curvature matrices explicitly.

We construct the preservation signal $\gamma$ from the diagonal Fisher $F_A^D$, which measures per-parameter sensitivity to Task A's loss. As this signal propagates through the network dynamics, forward through the feedforward pathway and backward through the learning signal $\psi$, parameter interactions emerge naturally. The interactions between parameters are now mediated by the modulated neural activities. At equilibrium, these interactions are encoded in how strongly

---

[1]Code available at https://github.com/s-de-haan/Equilibrium-Fisher-Control.

$\psi_\star$ must work against $\gamma$ to reach a loss-minimizing state. The off-diagonal entries of the full Fisher therefore arise implicitly from solving the network dynamics, without ever being explicitly computed or stored.

We formalize this observation as the continual-natural gradient property. Whereas the natural gradient preconditions parameter updates by the Fisher information of the current task thereby accounting for local curvature of the loss landscape (Amari, 1998), the continual-natural gradient preconditions by the previous task's curvature. Learning therefore proceeds within the geometry of already-acquired knowledge.

**Theorem 3.1.** *(Informal) For a small learning rate $\eta$ and linearization around $\theta_A^*$, the weight update for a sample $x_B$ satisfies:*

$$\Delta\theta \approx -\eta \tilde{F}_A^{-1}\nabla_\theta \mathcal{L}_B(x_B), \qquad (5)$$

*where $\tilde{F}_A \triangleq F_A\big|_{x_B}$ is an implicit approximation of the full Fisher information matrix $F_A$ that emerges from the network dynamics when processing sample $x_B$. An explicit form is derived in Supplementary C.*

This property distinguishes EFC from methods that store curvature information explicitly. We store only the diagonal Fisher in the preservation signal $\gamma$, yet recover an approximation to the full Fisher through the network dynamics themselves. Intuitively, the control minimization objective acts as a budget that the learning signal must allocate selectively. Because Task A's parameter sensitivities are embedded in the dynamics through $\gamma$, directions that conflict with Task A's curvature become expensive. The inversion is a direct result of our multiplicative modulation structure and the dynamical inversion property from Podlaski & Machens (2020). The learning signal therefore preferentially updates parameters orthogonal to Task A's loss landscape, precisely the behavior induced by preconditioning with $F_A^{-1}$. We verify empirically that $\tilde{F}_A$ captures the structure of $F_A$ in Section 4.1.

## 3.2. Class-incremental convergence

We now show that the continual-natural gradient property enables task discrimination, allowing the network to distinguish between classes learned at different times. This setting is known as class-incremental learning (van de Ven & Tolias, 2019), and is precisely where classical regularization methods fail regardless of the quality of their curvature approximation (de Lange et al., 2019; Masana et al., 2023).

We consider a classification problem with classes $\mathcal{C} = \mathcal{C}_A \cup \mathcal{C}_B$, where Task A classes $\mathcal{C}_A$ have been learned previously and Task B classes $\mathcal{C}_B$ are learned without access to Task A data. In class-incremental learning, all classes share a single output head over the combined class space, and no task identifier is provided at test time (van de Ven & Tolias, 2019). The gradient with respect to a Task B sample $x_B$

decomposes as

$$\nabla_\theta \mathcal{L}(x_B) = G_B(x_B) + G_{A \leftarrow B}(x_B) \qquad (6)$$

where $G_B$ drives learning of Task B and $G_{A \leftarrow B}$ arises from the coupling between classes in the shared output head (Supplementary D.1 derives this decomposition for cross-entropy and mean-squared error losses). Following the negative gradient improves Task B performance while suppressing Task A outputs as a direct consequence of the shared output head, a form of gradient interference studied in transfer and continual learning settings (Riemer et al., 2019).

### 3.2.1. WHY PARAMETER-BASED REGULARIZATION CANNOT CANCEL THE INTERFERENCE TERM

Parameter-based regularization methods modify training on Task B by adding a penalty $\mathcal{R}(\theta)$ that depends only on the current parameters. The resulting update adds the gradient $\nabla_\theta \mathcal{R}(\theta)$ to the Task B gradient. This modification does not alter how the Task B gradient itself is formed. In particular, the decomposition $G_B(x_B) + G_{A \leftarrow B}(x_B)$ is computed before the regularizer is applied, and both terms are passed unchanged to the optimizer.

The interference term $G_{A \leftarrow B}(x_B)$ depends on the activations induced by the specific input $x_B$. Different Task B samples therefore induce different interference directions. A parameter-based regularizer produces a single vector $\nabla_\theta \mathcal{R}(\theta)$ at a given parameter value and *cannot* cancel a sample-dependent quantity across all inputs. The regularizer can only counteract the expected interference through the stationary condition $\mathbb{E}[\nabla_\theta \mathcal{L}_B] + \nabla_\theta \mathcal{R} = 0$, while per-sample interference continues to vary around this expectation with variance $\text{Var}(G_{A \leftarrow B})$, suppressing Task A outputs.

This limitation is independent of the quality of the curvature approximation encoded by the regularizer. Even when curvature information from Task A is available, the regularizer contributes an additive correction after task-relevant and interfering components have already been combined in the gradient. Attenuation therefore applies to the combined signal rather than selectively to $G_{A \leftarrow B}$. A full derivation and formal analysis of this failure mode are provided in Supplementary D.2.

### 3.2.2. EFC FILTERS THE INTERFERENCE TERM

We now show that EFC filters the interference term from the gradient, thereby enabling task discrimination. We denote by $\mathcal{V}_A$ the subspace of parameter directions along which Task A's loss is sensitive, defined as the column space of the Task A Fisher information matrix (Supplementary D.3). The interference term $G_{A \leftarrow B}$ lies in $\mathcal{V}_A$ by construction. The EFC update from Theorem 3.1 preconditions by $\tilde{F}_A^{-1}$, which has small eigenvalues along directions in $\mathcal{V}_A$ and

large eigenvalues along directions orthogonal to $\mathcal{V}_A$. Components of the gradient lying in $\mathcal{V}_A$ are therefore attenuated, while components orthogonal to $\mathcal{V}_A$ pass through with less attenuation. This filtering occurs automatically through the structure of the update, without requiring explicit identification of $G_{A \leftarrow B}$.

**Theorem 3.2.** *Let $\tilde{\lambda}_{min}$ denote the minimum eigenvalue of $\tilde{F}_A$ restricted to $\mathcal{V}_A$. Decomposing $G_B = G_B^\parallel + G_B^\perp$ into components parallel and orthogonal to $\mathcal{V}_A$, the EFC update satisfies:*

$$\Delta\theta_{EFC} \propto \tilde{F}_A^{-1} G_B^\perp + O(\tilde{\lambda}_{min}^{-1}) \qquad (7)$$

*thereby guaranteeing convergence on the class-incremental objective,*

$$\mathcal{L}^{CIL}(\theta + \Delta\theta_{EFC}) - \mathcal{L}^{CIL}(\theta) \leq -\eta \|G_B^\perp\|_{\tilde{F}_A^{-1}}^2 + O(\tilde{\lambda}_{min}^{-1}) \quad (8)$$

Learning therefore proceeds in directions orthogonal to Task A's curvature, while both $G_{A \leftarrow B}$ and $G_B^\parallel$ are suppressed by factor $\tilde{\lambda}_{\min}^{-1}$. When $\tilde{\lambda}_{\min}$ is large, each update guarantees descent on the class-incremental objective. The proof follows from the continual-natural gradient property and the spectral structure of $\tilde{F}_A^{-1}$ on $\mathcal{V}_A$ (Supplementary D.5).

EFC thus requires no explicit knowledge of which gradient components are harmful. Interfering with Task A necessarily requires modifying Task A-sensitive parameters, and the continual-natural gradient attenuates exactly these directions. We verify convergence empirically in Section 4.1.

### 3.3. Forgetting bounds in task-incremental learning

We now show that EFC tightens forgetting bounds compared to regularization-based methods in task-incremental learning, the setting where task identifiers are known at test time and no cross-task confusion can arise (van de Ven & Tolias, 2019). We quantify forgetting directly as the increase in Task A's loss caused by parameter updates during Task B training. We compare standard backpropagation (BP), EWC as the canonical regularization-based method, and EFC. Our starting point is the Laplace approximation around any parameter change $\Delta\theta = \theta - \theta_A^*$,

$$\mathcal{L}_A(\theta_A^* + \Delta\theta) - \mathcal{L}_A(\theta_A^*) \approx \tfrac{1}{2}\Delta\theta^\top F_A \Delta\theta \qquad (9)$$

where the forgetting depends on how each method's parameter deviation interacts with Task A's curvature $F_A$. The three methods induce different parameter updates,

$$\Delta\theta_{\text{BP}} \propto \nabla_\theta \mathcal{L}_B \qquad (10)$$

$$\Delta\theta_{\text{EWC}} \propto D_A^{-1} \nabla_\theta \mathcal{L}_B \qquad (11)$$

$$\Delta\theta_{\text{EFC}} \propto \tilde{F}_A^{-1} \nabla_\theta \mathcal{L}_B \qquad (12)$$

where $D_A = \text{diag}(F_A)$ is the diagonal Fisher used by EWC. The EWC update follows from linearizing around $\theta_A^*$ when

regularization strength dominates the Task B Hessian. We substitute these deviations into Equation 9 to obtain the following bounds.

**Theorem 3.3.** *Let $\lambda_{\max}$ denote the maximum eigenvalue of $F_A$, let $\lambda_{\min}^D$ denote the minimum eigenvalue of $D_A$, and let $\tilde{\lambda}_{\max}$ denote the maximum eigenvalue of $\tilde{F}_A$. Define $\Delta \mathcal{L}_A \triangleq \mathcal{L}_A(\theta_A^* + \Delta\theta) - \mathcal{L}_A(\theta_A^*)$. The forgetting bounds satisfy*

$$\Delta \mathcal{L}_A^{BP} \lesssim \|\nabla_\theta \mathcal{L}_B\|^2 \lambda_{\max} \tag{13}$$

$$\Delta \mathcal{L}_A^{EWC} \lesssim \|\nabla_\theta \mathcal{L}_B\|^2 \frac{n}{\lambda_{\min}^D} \tag{14}$$

$$+ O(\|\nabla_\theta^2 \mathcal{L}_B\|) + O(\|D_A - F_A\|) \tag{15}$$

$$\Delta \mathcal{L}_A^{EFC} \lesssim \|\nabla_\theta \mathcal{L}_B\|^2 \tilde{\lambda}_{\max}^{-1} + O(\|\tilde{F}_A - F_A\|) \tag{16}$$

*where $n$ is the number of parameters and $\lesssim$ absorbs constant factors involving $\eta^2$. The formal statement and proof are in Supplementary E.2. Pairwise comparisons are in Supplementary E.3.*

The bounds differ in how forgetting scales with problem structure. BP lacks any mechanism to account for Task A's curvature, giving the loosest bound. EWC tightens the bound by incorporating the diagonal Fisher, but the bound scales with the number of parameters $n$ through $\lambda_{\min}^D$. This scaling arises because the diagonal approximation discards interactions between parameters, treating each parameter independently. EFC obtains the tightest bound by implicitly capturing off-diagonal curvature through $\tilde{F}_A$, avoiding the dimensional scaling entirely.

We draw a second distinction concerning the coupling with Task B's curvature. The EWC bound includes a term $O(\|\nabla_\theta^2 \mathcal{L}_B\|)$ that reflects a structural limitation of parameter-based regularization. At the stationary point of the regularized objective, the regularizer $D_A$ couples additively with Task B's Hessian through $\nabla_\theta^2 \mathcal{L}_B + \beta D_A = 0$. This coupling persists even when replacing $D_A$ with the full Fisher $F_A$. As we have seen, EFC avoids this coupling because Task A's curvature enters as a preconditioner rather than as an additive penalty.

We can interpret the difference between EWC and EFC geometrically. EWC constrains parameters to a hypercube aligned with parameter axes, while EFC constrains parameters to a hyperellipsoid aligned with Task A's curvature. The volume ratio between these regions grows exponentially with dimension when $F_A$ has significant off-diagonal structure (Supplementary E.4). The sample-specific nature of EFC provides an additional advantage. As discussed in Section 3.2, regularization can only counteract expected interference while per-sample deviations persist. Because EFC computes preservation on each sample, the expected forgetting bound is further tightened by the variance in preservation strength across samples (Supplementary E.5).

## 4. Experiments

Here, we validate the theoretical predictions from Section 3 and evaluate EFC on standard continual learning benchmarks. We first use a small, controlled setup that permits exact computation of the full Fisher while retaining the challenges of continual learning in Section 4.1, then compare EFC against existing methods on Split-MNIST, Split-CIFAR10, and Split-Tiny-ImageNet in Section 4.2.

### 4.1. Empirical validation of the learning theory

In order to computationally permit calculating the full Fisher, we train a two-hidden-layer MLP with $64$ neurons per layer on two-task Split-MNIST in the class-incremental setting, using learning rate $0.0001$ and the Adam optimizer (Kingma & Ba, 2014). We assume Task A has been learned and optimal parameters $\theta_A^*$ have been obtained, after which we train on Task B.

We compare four methods: standard backpropagation (BP), EWC, a variant of EWC that replaces the diagonal Fisher with the full Fisher to represent optimal regularization (FISH), and our Equilibrium Fisher Control (EFC) framework. We ensure comparability by starting all methods from the same parameters $\theta_A^*$ and aligning training so that each method reaches $90\%$ accuracy on Task B after the same number of training steps.

#### 4.1.1. The continual-natural gradient approximates the full Fisher

We first verify the quality of the implicit curvature approximation from Theorem 3.1 by measuring the increase in Task A loss during Task B training (Figure 1a). The forgetting for each method reflects how well its curvature approximation $M$ satisfies $\|M - F_A\|$, with $M = I$ for BP, $M = D_A$ for EWC, $M = \tilde{F}_A$ for EFC, and $M = F_A$ for FISH.

BP exhibits the highest forgetting, consistent with the identity matrix being a poor approximation of the full curvature. The normalized Frobenius inner product $\langle I, F_A \rangle / (\|I\|\|F_A\|) \approx 0.01$ confirms this quantitatively. EWC reduces forgetting relative to BP, though the diagonal approximation remains limited, with $\langle D_A, F_A \rangle / (\|D_A\|\|F_A\|) \approx 0.09$. EFC matches the forgetting profile of FISH, confirming that the dynamical system recovers an approximation to the full Fisher despite storing only diagonal information in the preservation signal $\gamma$.

#### 4.1.2. EFC achieves class-incremental convergence

We next verify that EFC converges on the class-incremental objective where backpropagation with regularization does not (Figure 1b). All backpropagation-based methods, in-

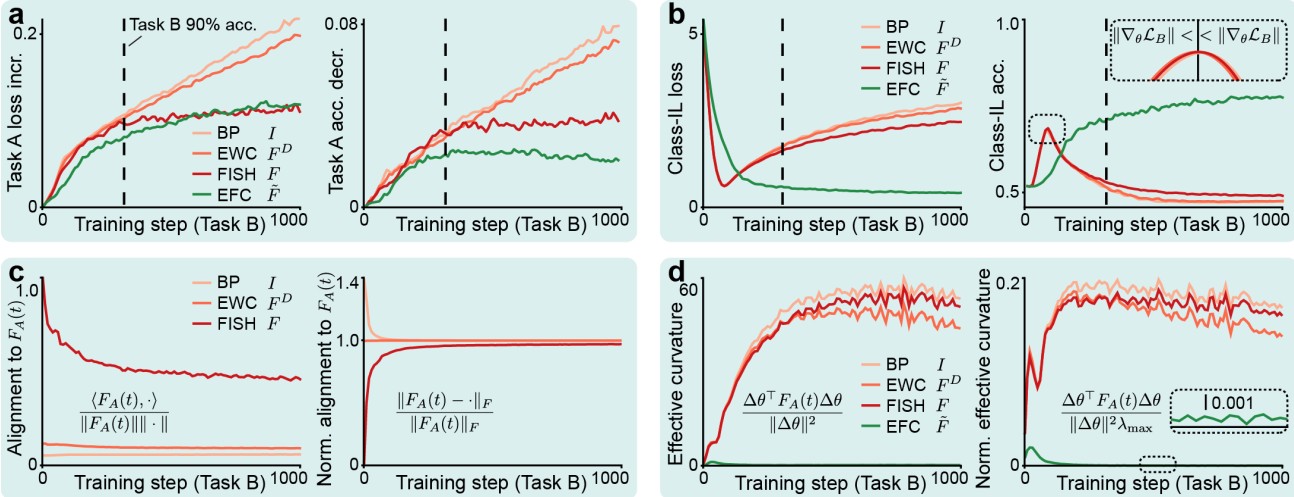

*Figure 1.* Empirical validation of the learning theory. (a) Task A loss increase (left) and accuracy decrease (right) during Task B training; the dynamical approximation tracks the full Fisher, and EFC retains higher accuracy. (b) Class-IL loss (left) and accuracy (right); backpropagation-based methods fail to converge while EFC converges. (c) Progressive misalignment between the true current Fisher and three static approximations, normalized Frobenius inner product (left) and distance (right). EFC is absent as its curvature approximation is implicit. (d) Effective curvature of each method's updates measured as the Rayleigh quotient against the true local Fisher (left), and normalized by the largest eigenvalue (right). EFC operates near zero, confirming that its dynamics avoid Task A's sensitive directions despite never storing curvature explicitly.

cluding FISH with access to the full curvature $F_A$, fail to converge on the class-incremental loss. The loss initially decreases as Task B learning dominates, reaches a stationary point, and subsequently increases as the accumulated interference term $G_{A \leftarrow B}$ overtakes learning progress. This pattern matches the theoretical prediction from Section 3.2: parameter-based regularization cannot cancel the sample-dependent interference term.

As we predicted by Theorem 3.2, EFC converges on the class-incremental loss. The preservation signal $\gamma$ modulates the dynamics on a per-sample basis, filtering the interference term before parameter updates occur. The divergence of backpropagation-based methods after the stationary point reflects the growing influence of Task B's curvature in the weights, analyzed in detail in Supplementary F.

### 4.1.3. DYNAMIC CURVATURE ESTIMATION OUTPERFORMS STATIC STORAGE

We observe that Task A loss increases similarly for EFC and FISH, while Task A accuracy degrades less for EFC (Figure 1a, right panel). We will show that this reflects a limitation of storing curvature information explicitly, even when the full Fisher is available, instead of dynamically obtaining the curvature information.

We hypothesize that the Fisher computed at $\theta_A^*$ characterizes the local geometry at that specific point in parameter space. As parameters move during Task B training, this stored curvature becomes increasingly misaligned with the true local geometry. FISH anchors the full Fisher at $\theta_A^*$ and cannot

adapt. EFC stores only the diagonal Fisher in the preservation signal $\gamma$, while the off-diagonal structure emerges dynamically from the network's feedforward-feedback interactions at the current parameter configuration. The implicit curvature $\tilde{F}_A$ therefore transforms appropriately as parameters move, remaining consistent with the geometry of the combined Task A and Task B landscape.

We quantify this progressive misalignment directly by computing the true Fisher $F_A(t)$ at the current parameters $\theta$ at each training step $t$ Task A data over the full 10-class output head, then measuring its alignment with three static references computed once at $\theta_A^*$ (Figure 1c). The normalized Frobenius inner product shows that the stored full Fisher loses approximately 15% of its structural alignment within one epoch and 50% within ten epochs. The normalized Frobenius distance confirms this, with all three static approximations converging toward a relative distance of 1.0 as training progresses. Even the best possible static approximation therefore becomes increasingly uninformative about the true local geometry.

Since $\tilde{F}_A$ is never constructed explicitly, we measure its consequences instead by asking whether each method's parameter updates avoid Task A's sensitive directions (Figure 1d). For each method following its own training trajectory, we compute the Rayleigh quotient of the cumulative parameter displacement with respect to the true current Fisher evaluated at that method's own parameters, measuring the effective curvature experienced per unit of parameter movement. EFC maintains effective curvature near zero through-

| | Method | | Split-MNIST | | Split-CIFAR10 | | Split-Tiny-ImageNet | |
|---|---|---|---|---|---|---|---|---|
| | | | Task-IL | Class-IL | Task-IL | Class-IL | Task-IL | Class-IL |
| **Baselines** | JOINT | (upper bound) | 99.7±0.0 | 98.2±0.1 | 98.3±0.1 | 92.2±0.2 | 82.0±0.1 | 60.0±0.2 |
| | SGD | (lower bound) | 98.1±0.1 | 19.3±0.0 | 95.5±0.5 | 19.6±0.1 | 28.0±0.5 | 3.8±0.1 |
| **Parameter-based regularization** | EWC | (Kirkpatrick et al., 2017) | 97.2±0.6 | 19.8±0.0 | 95.5±1.2 | 20.9±1.1 | 34.2±0.4 | 4.6±0.1 |
| | oEWC | (Schwarz et al., 2018) | **99.2±0.4** | 20.2±0.8 | 95.9±0.7 | 21.2±2.0 | 35.5±0.3 | 4.8±0.0 |
| | SI | (Zenke et al., 2017) | 97.5±0.9 | 19.7±0.0 | 92.6±1.3 | 19.8±0.1 | 35.5±0.5 | 4.7±0.0 |
| | CSQN | (Eeckt & Hamme, 2025) | 98.2±0.3 | 19.2±0.0 | 95.4±0.7 | 20.2±1.2 | 35.5±0.5 | 4.8±0.1 |
| | **EFC** | **(ours)** | 98.3±0.7 | **51.4±4.8** | **96.2±0.3** | **50.2±7.0** | **37.2±0.4** | **8.8±0.1** |
| **Replay (200)** | DER++ | (Buzzega et al., 2020) | 98.6±0.2 | 71.1±2.2 | 95.4±0.8 | 62.3±1.2 | 39.0±1.6 | 10.5±1.2 |

*Table 1.* Classification accuracies on continual learning benchmarks. We report mean ± standard deviation over five random seeds. JOINT trains on all tasks simultaneously (upper bound). SGD trains sequentially without any continual learning mechanism (lower bound).

out training, while BP, EWC, and FISH cluster three orders of magnitude higher. Normalizing by the largest eigenvalue $\lambda_{\max}$ of $F_A(t)$ bounds the quantity in $[0, 1]$, where zero represents movement perfectly orthogonal to Task A's curvature. EFC again operates near zero, confirming that its dynamics steer parameter updates away from Task A's sensitive subspace as that subspace evolves.

### 4.2. Benchmarks

We evaluate EFC against standard parameter-based regularization methods on Split-MNIST, Split-CIFAR10, and Split-Tiny-ImageNet across task-incremental and class-incremental settings (van de Ven & Tolias, 2019). We compare against Elastic Weight Consolidation (EWC; Kirkpatrick et al., 2017), online EWC (oEWC; Schwarz et al., 2018), and Synaptic Intelligence (SI; Zenke et al., 2017) as representative parameter-based regularization methods. We additionally evaluate Dark Experience Replay (DER++; Buzzega et al., 2020) with a buffer size of 200 samples, to quantify the gap between regularization and replay approaches.

#### 4.2.1. ARCHITECTURES AND TRAINING PROTOCOL

For Split-MNIST, we follow Riemer et al. (2019) and train a fully-connected network with two hidden layers of 100 ReLU units each, without pretraining. For Split-CIFAR10 and Split-Tiny-ImageNet, we use a pretrained ResNet18 (He et al., 2016) as a frozen encoder and train a fully-connected layer of width 100 for Split-CIFAR10 and 1000 for Split-Tiny-ImageNet, followed by the output layer. We chose to use larger network size for Split-Tiny-ImageNet due to the significantly larger number of classes (200 for Tiny-ImageNet as opposed to 10 for CIFAR-10 and MNIST). We use a pretrained encoder to isolate the failure mode of parameter-based regularization in class-incremental learning, as established in Section 3.2.1, demonstrating that the failure persists even when the learned features are sufficient for the task, while not diminishing the core difficulty of

continual learning. We apply no data augmentation. Each method is hyperparameter-optimized individually, and we refer the reader to our codebase for a per-method and per-setting list. All methods use the Adam optimizer (Kingma & Ba, 2014) except the backpropagation SGD baseline, which serves as a lower bound. We train each task for 20 epochs. We evaluate each method over five random seeds and report mean accuracy with standard deviation.

#### 4.2.2. EXPERIMENTAL RESULTS

We gather our experimental results in Table 1. EFC consistently matches parameter-based regularization methods across all settings, with particularly pronounced improvements in class-incremental learning. Buzzega et al. (2020) have described an "unbridgeable gap" between regularization methods and replay-based approaches in class-incremental settings. Our results substantially narrow this gap without relying on replay.

Prior work attributed the failure of regularization methods in class-incremental settings to the local nature of curvature estimates, arguing that importance computed at earlier task optima becomes unreliable as parameters move through later tasks (Buzzega et al., 2020; Kim et al., 2022; Masana et al., 2023; Wu et al., 2024). Our framework addresses this limitation through dynamic curvature estimation, where the network dynamics approximate prior-task curvature at the current parameter configuration rather than relying on stored estimates. However, our theoretical analysis in Section 3.2.1 identifies a more fundamental obstacle. Parameter-based regularization cannot distinguish gradient components that help the new task from components that interfere with previous tasks, regardless of curvature quality. The experimental results confirm this analysis. Methods with improved curvature approximations fail to improve in class-incremental settings, whereas EFC achieves substantial gains by filtering interference within the dynamics before parameter updates occur.

We additionally visualize this phenomenon through confusion matrices for oEWC, DER++, and EFC in Figure S1. Regularization methods shift nearly all output probability mass to the most recently learned task, with earlier tasks receiving close to zero accuracy. This pattern confirms our theoretical prediction that parameter-based regularization cannot prevent the suppression of prior-task outputs during new-task learning. DER++ maintains more balanced accuracy across tasks, with confusion occurring primarily between the current sample's class and classes from previously learned tasks. EFC exhibits bidirectional confusion, misclassifying samples both as earlier and later task classes, indicating that preservation and learning signals interact throughout the task sequence rather than simply favoring recent tasks (see Supplementary G).

## 5. Discussion

Our work reformulates continual learning as a control problem where learning and preservation signals compete within neural activity dynamics. By converting regularization penalties into neuron-specific preservation signals that modulate network dynamics, we couple the learning of new tasks directly to the preservation of prior tasks during the formation of learning signals. In contrast, parameter-based regularization methods fail in settings requiring task discrimination because corrections are applied after gradient computation, leaving new-task learning decoupled from prior-task geometry. Our framework addresses both limitations of existing regularization methods identified in our analysis. Curvature is approximated dynamically rather than stored statically, and interference is filtered within the dynamics rather than corrected after the gradient has already been computed. At equilibrium, the resulting weight updates implicitly encode the full prior-task curvature without explicit storage, a property we term the continual-natural gradient. Our experiments validate these theoretical predictions and demonstrate improved performance over existing replay-free methods on standard benchmarks.

Our learning framework draws inspiration from the computational properties of cortical pyramidal neurons. We model our neurons after layer 5 pyramidal neurons, which possess a compartmentalized dendritic architecture where basal dendrites integrate feedforward sensory information and apical dendrites receive diverse top-down signals, including feedback from higher cortical regions, teaching signals, and contextual information that guides both learning and preservation (Spruston, 2008; Larkum et al., 2009; Larkum, 2013; Williams & Holtmaat, 2019). Three properties of pyramidal neurons inform our framework. First, apical inputs act multiplicatively on neuronal output rather than additively, allowing apical signals to modulate the feedforward processing (Larkum et al., 2004; Aceituno et al., 2025). Our

learning and preservation signals employ this multiplicative structure, modulating neural activities proportionally to how strongly current inputs engage previously relevant synapses. Second, apical inputs are co-directional with basal synaptic plasticity, meaning that apical modulation guides the direction of synaptic weight changes (Payeur et al., 2021; Aceituno et al., 2025). Our framework applies this principle by deriving weight updates from equilibrium states shaped by the interaction of learning and preservation signals, thereby also being co-directional in modulation and plasticity. Third, cortical networks operate as dynamical systems where information flows bidirectionally across the hierarchy (Felleman & Van Essen, 1991; Bastos et al., 2012), and learning proceeds by driving neural activities toward target states rather than propagating explicit errors backward (Meulemans et al., 2021; 2022b; Song et al., 2024). This target learning perspective aligns with our least-control formulation, where parameters update to make desired equilibria easier to reach. Our work therefore contributes to a growing body of research on biologically plausible learning algorithms (Whittington & Bogacz, 2017; Guerguiev et al., 2017; Scellier & Bengio, 2017; Richards et al., 2019; Lillicrap et al., 2020; Meulemans et al., 2021; Song et al., 2024; Aceituno et al., 2025) by demonstrating that principles derived from cortical computation can address fundamental challenges in continual learning.

We view the current work as a proof-of-principle that continual learning can be reformulated as a control minimization problem and approached through a dynamical systems lens, but several limitations warrant discussion. First, dynamical systems approaches are substantially more computationally expensive than standard backpropagation, as each learning step requires running network dynamics to equilibrium through multiple iterations of forward and feedback passes. The per-iteration cost scales linearly with the number of trainable neurons, and convergence speed depends on the conditioning of the network Jacobian, which tends to worsen with depth, limiting applicability to problems where control-based methods remain tractable. Our current experiments train moderate-sized MLPs, either from scratch or on top of frozen encoders, and a gap remains between our evaluation and large-scale end-to-end continual learning on deep architectures. The broader utility of this work is therefore currently in establishing that control-based learning can address failure modes that no parameter-based regularization method can resolve, rather than in immediate practical deployment. Bridging this gap will require reducing the cost of running dynamics to equilibrium, which we consider the primary open challenge for control-based approaches to learning. Second, our framework introduces additional hyperparameters governing the dynamics, including the preservation strength, controller leak rate, and different convergence criteria. We provide a systematic analysis of

these hyperparameters in Supplementary H. Combined with standard optimization hyperparameters, this increases the complexity of training and makes hyperparameter tuning more tedious. The sensitivity of dynamical systems to these settings can lead to instability or slow convergence when hyperparameters are not carefully chosen. Third, while the continual-natural gradient captures parameter interactions dynamically, our framework still requires storing the diagonal of the Fisher information matrix to construct the preservation signal. This stored curvature suffers from the same drift problem identified for other regularization methods, namely as parameters move during sequential learning, the stored diagonal becomes increasingly misaligned with the true local geometry. Our framework mitigates but does not eliminate this limitation, as the off-diagonal structure is recovered dynamically while the diagonal remains static.

The limitation of storing diagonal curvature points toward a natural direction for future work. A fully dynamical approach would approximate all relevant curvature information through network dynamics without storing any curvature explicitly. Such methods could leverage the sample-specific properties demonstrated in this work while eliminating the mismatch between stored and true curvature entirely. One avenue is to derive preservation signals from activity-based importance measures that evolve online during learning, rather than from curvature computed at previous optima. Another direction involves extending the least-control framework to settings without explicit task boundaries, where preservation must operate continuously rather than being triggered by task transitions. The connection between our framework and biological learning mechanisms also suggests experimental predictions, namely if cortical networks implement something analogous to our preservation signal, neurons encoding task-relevant representations should exhibit increased resistance to modulation during subsequent learning, a prediction testable through longitudinal imaging of neural populations across sequential task acquisition.

## Contributions

S.d.H. conceptualized the project, performed the mathematical analyses, and implemented the framework. S.d.H. and Y.T.B. performed the experiments. P.V.A. and B.F.G. supervised the project. S.d.H., Y.T.B., P.V.A, and B.F.G. wrote the paper.

## Acknowledgements

This work was supported by the Swiss National Science Foundation (B.F.G. CRSII5-173721 and 315230 189251, P.V.A. 182539funding), ETH project funding (B.F.G. ETH-20 19-01), a Forschungskredit from the University of Zürich (P.V.A. FK-24-122), and part of the "Learn to learn safely" project funded by a grant of the Hasler foundation (Y.T.B. 21039). We want to especially thank Joonsu Gha for support on the codebase and project, and Giulia Lanzillotta for the great discussions that led to the conceptualization of the project.

## Impact Statement

This paper presents work whose goal is to advance the field of Machine Learning. There are many potential societal consequences of our work, none which we feel must be specifically highlighted here.

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

# Notation Guide

### Network and parameters

| | |
|---|---|
| $L$ | Number of layers |
| $N$ | Total number of neurons across all layers |
| $d_i$ | Dimension (number of neurons) of layer $i$ |
| $\theta$ | Parameters; $\theta_k$ denotes parameters incoming to neuron $k$ |
| $\theta_A^*$ | Optimal parameters after training on Task A |
| $W_i \in \mathbb{R}^{d_i \times d_{i-1}}$ | Weight matrix from layer $i-1$ to layer $i$ |
| $f(\phi, \theta)$ | Mapping of the network |

### Neural activities and dynamics

| | |
|---|---|
| $\phi \in \mathbb{R}^N$ | Concatenated neural activities across the whole network; $\phi_\star$ at equilibrium |
| $\mathbf{r}_i \in \mathbb{R}^{d_i}$ | Post-nonlinearity activities of layer $i$; $\mathbf{r}_0$ is the fixed input; $\mathbf{r}_\star$ at equilibrium |
| $\mathbf{r}_i^-$ | Feedforward (uncontrolled) activities: $\mathbf{r}_i^- = \sigma(W_i \mathbf{r}_{i-1}^-)$ |

### Control and preservation signals

| | |
|---|---|
| $\psi \in \mathbb{R}^N$ | Learning (control) signal; $\psi_k$ is the signal for neuron $k$ |
| $\psi_\star$ | Learning signal at equilibrium |
| $\gamma \in \mathbb{R}^N$ | Preservation signal; $\gamma_k$ is the signal for neuron $k$ |
| $\mathbf{u}(t)$ | Controller state; drives $\psi$ via feedback weights $Q$ |
| $Q_i$ | Feedback weight matrix mapping $\mathbf{u}$ to layer $i$'s learning signal |
| $\alpha$ | Controller leak rate (damping coefficient) |
| $\beta > 0$ | Preservation strength hyperparameter |

### Curvature and Fisher information

| | |
|---|---|
| $F_A$ | Full Fisher information matrix of Task A at $\theta_A^*$ |
| $D_A \triangleq F_A^D$ | Diagonal of $F_A$; $F_{A,k}^D$ restricts to neuron $k$'s parameters |
| $\tilde{F}_A$ | Implicit (dynamical) approximation of $F_A$ arising from EFC |
| $H_A$ | Hessian of Task A's loss at $\theta_A^*$ |
| $\mathcal{V}_A$ | Task A-sensitive subspace: $\mathcal{V}_A = \text{col}(F_A)$ |

### Gradient decomposition (class-incremental)

| | |
|---|---|
| $\mathcal{C}_A, \mathcal{C}_B$ | Class sets for Task A and Task B |
| $G_B(x_B)$ | Task B learning gradient component |
| $G_{A \leftarrow B}(x_B)$ | Interference gradient from shared output head |
| $G_B^{\parallel}, G_B^{\perp}$ | Components of $G_B$ parallel and orthogonal to $\mathcal{V}_A$ |

### Jacobians and steady-state quantities

| | |
|---|---|
| $J_{i,i-1}$ | Local Jacobian: $\sigma'(W_i \mathbf{r}_{i-1}^-) \odot W_i$ |
| $J_i$ | Cumulative Jacobian from layer $i$ to output: $\prod_{k=i+1}^{L} J_{k,k-1}$ |
| $J$ | Block lower-triangular Jacobian matrix of the full network |
| $J_{\text{eff}}$ | Effective Jacobian from control to output (Theorem B.1) |
| $\gamma_{\text{eff}}$ | Effective preservation signal propagated to output (Theorem B.1) |

# A. Multiplicative least-control

## A.1. Deriving the preservation signal

Continual learning seeks parameters $\theta$ that fit a joint data distribution $\mathcal{D} = \{\mathcal{D}_A, \mathcal{D}_B\}$, where Task A has been learned previously, and Task B is now trained without further access to $\mathcal{D}_A$. From a Bayesian perspective, this is framed as maximizing the posterior:

$$\log p(\theta|\mathcal{D}_A, \mathcal{D}_B) = \underbrace{\log p(\mathcal{D}_B|\theta)}_{-\mathcal{L}_B(\theta)} + \underbrace{\log p(\theta|\mathcal{D}_A)}_{\text{crucial for CL}} - \underbrace{\log p(\mathcal{D}_B)}_{\text{independent of } \theta} \tag{S1}$$

This decomposes into optimizing $\theta$ for Task B (via the likelihood) while retaining Task A's knowledge (via the posterior). The difficulty lies in preserving $p(\theta|\mathcal{D}_A)$ without having access to $\mathcal{D}_A$. We follow the standard Laplace approximation from Kirkpatrick et al. (Kirkpatrick et al., 2017) around the optimal parameters $\theta_A^*$ from Task A:

$$\log p(\theta|\mathcal{D}_A) \approx \log p(\theta_A^*|\mathcal{D}_A) - \frac{1}{2}(\theta - \theta_A^*)^\top H_A(\theta - \theta_A^*) \tag{S2}$$

where $H_A$ is the Hessian of Task A's loss at $\theta_A^*$ and a Gaussian posterior is assumed. The gradient of this posterior, guiding preservation during Task B training, is,

$$\nabla_\theta \log p(\theta|\mathcal{D}_A) = -H_A(\theta - \theta_A^*) \overset{(1)}{\approx} -F_A(\theta - \theta_A^*) \overset{(2)}{\approx} -F_A^D(\theta - \theta_A^*) \tag{S3}$$

Two practical approximations are ubiquitous in the literature and preserved here: (1) $H_A \approx F_A$, the Fisher information matrix evaluated at $\theta_A^*$ (exact for standard supervised losses at a strict local minimum), and (2) the diagonal Fisher $F_A^D$ for scalability, as the full Fisher information matrix requires $O(\theta^2)$ computations, yielding the classic EWC penalty (Kirkpatrick et al., 2017).

Now as an intermezzo, any parameter-space objective $\mathcal{R}(\theta)$, think of a regularizer or Bayesian prior, can be turned into a neuron-specific constraint signal. We simply project the relevant gradient block onto the current presynaptic activities of layer $i$ and neuron $k$,

$$\gamma_{i,k} = \beta\, \mathbf{r}_{i-1,\star}^\top \nabla_{W_i[:,k]} \mathcal{R}(\theta) \tag{S4}$$

where $W_i[:,j]$ denotes the incoming weight column for neuron $j$, and $\beta \in \mathbb{R}^+$ scales strength. The resulting constraint is computed solely from local presynaptic rates and parameters, making the signal fully neuron-specific and activity-dependent. Consolidation or amplification occurs only when the synapses that are simultaneously important to $\mathcal{R}$ and currently active.

We now apply this to the EWC regularizer, therefore the resulting preservation constraint, which we denote as $\gamma$, is (in simplified vector form),

$$\gamma_i = \beta\, \mathbf{r}_{i-1,\star}^\top F_{A,i}^D(W_i - W_{A,i}^*),$$

with $\beta > 0$ controlling preservation strength. We therefore consider the following multiplicative constrained least control objective for continual learning,

$$\min_{\theta,\phi,\psi} \|\psi\|^2 \quad \text{s.t.} \quad e^{\psi+\gamma} \odot f(\phi,\theta) = \mathbf{1} \ \wedge \nabla_\phi \mathcal{L}(\phi) = \mathbf{0} \tag{S5}$$

## A.2. Definition

Previous work (Meulemans et al., 2022b) defines a least-control principle, reframing optimization away from backpropagation and towards control theory. In brief, the normal objective of:

$$\min_\theta \mathcal{L}(\phi^*) \text{ s.t. } f(\phi^*, \theta) = 0, \text{ with } \dot\phi = f(\phi,\theta) \tag{S6}$$

where $\phi^*$ represents the dynamics at equilibrium, can be restructured by introducing a control signal and minimizing the amount of control required:

$$\dot\phi = f(\phi,\theta) + \psi \implies \min_{\theta,\phi,\psi} \underbrace{\|\psi\|^2}_{\text{mag. of control}} \quad \text{s.t.} \quad \underbrace{f(\phi,\theta) + \psi = 0}_{\text{steady state of } \phi} \wedge \underbrace{\nabla_\phi \mathcal{L}(\phi) = 0}_{\text{minima of dynamics}} \tag{S7}$$

The parameters are then updated by first fixing $\theta$ and finding the optimal $\phi_*$ and $\psi_*$ and then updating the weights with the gradient of $\mathcal{H}(\theta) := ||\psi||^2$.

We now extend this to a multiplicative variant with the objective

$$\dot{\phi} = f(\phi, \theta) + \psi \implies \min_{\theta, \phi, \psi} ||\psi||^2 \text{ s.t. } e^\psi f(\phi, \theta) - 1 = 0 \wedge \nabla_\phi \mathcal{L}(\phi) = 0 \tag{S8}$$

In this section we will proof the first-order gradient of the multiplicative least-control objective, followed by the general class of controllers that solve the objective together with a comparison between the additive case of (Meulemans et al., 2022b) and ours, and finally the convergence of the network together with the equilibrium points. Some of proofs follow a similar structure as explored by (Meulemans et al., 2022b).

### A.3. First-order gradient

**Theorem A.1** (First-order gradient). *Let $(\phi_\star, \psi_\star)$ be an optimal control for the multiplicative least-control problem of Eq. S8, and let $(\lambda_\star, \mu_\star)$ be the Lagrange multipliers for which the KKT conditions are satisfied. Under the assumption that the Hessian of the multiplicative LCP-Lagrangian $\partial^2_{\phi_\star, \psi_\star, \lambda_\star, \mu_\star, \theta}$ is invertible, the multiplicative LCP yields the following gradient for $\theta$:*

$$\left( \frac{d}{d\theta} \mathcal{H}(\theta) \right)^\top = \partial_\theta \log f(\phi_\star, \theta)^\top \log f(\phi_\star, \theta) \tag{S9}$$

*Proof.* We first state the KKT condition for the multiplicative LCP Lagrangian:

$$\mathcal{L}(\phi, \psi, \lambda, \mu, \theta) = \frac{1}{2} ||\psi||^2 + \lambda^\top e^\psi f(\phi, \theta) - 1 + \mu^\top \nabla_\phi \mathcal{L}(\phi) \tag{S10}$$

$$\partial_\phi \mathcal{L}(\phi_\star, \psi_\star, \lambda_\star, \mu_\star, \theta) = \lambda_\star^\top e^{\psi_\star} \partial_\phi f(\phi_\star, \theta) + \mu_\star^\top \partial_y^2 L(h(\phi_\star)) \partial_\phi h(\phi_\star) = 0 \tag{S11}$$

$$\partial_\psi \mathcal{L}(\phi_\star, \psi_\star, \lambda_\star, \mu_\star, \theta) = \psi_\star^\top + \lambda_\star^\top e^{\psi_\star} f(\phi_\star, \theta) = 0 \tag{S12}$$

$$\partial_\lambda \mathcal{L}(\phi_\star, \psi_\star, \lambda_\star, \mu_\star, \theta) = e^{\psi_\star} f(\phi_\star, \theta)^\top - 1 = 0 \tag{S13}$$

$$\partial_\mu \mathcal{L}(\phi_\star, \psi_\star, \lambda_\star, \mu_\star, \theta) = \partial_y L(h(\phi_\star)) = 0 \tag{S14}$$

now we have two additional constraints,

$$\psi_\star^\top + \lambda_\star^\top e^{\psi_\star} f(\phi_\star, \theta) = 0 \implies \lambda_\star^\top = -\frac{\psi_\star^\top}{e^{\psi_\star} f(\phi_\star, \theta)} \tag{S15}$$

$$e^{\psi_\star} f(\phi_\star, \theta)^\top - 1 = 0 \implies \psi_\star = -\log f(\phi_\star, \theta) \tag{S16}$$

with which we can prove our theorem, as $(\phi_\star, \psi_\star)$ are an optimal control, there exist $(\lambda_\star, \mu_\star)$ such that $\partial_{\phi, \psi, \lambda, \mu} \mathcal{L}(\phi_\star, \psi_\star, \lambda_\star, \mu_\star, \theta) = 0$ as is our assumption. We use this condition to implicitly define $(\phi_\star, \psi_\star, \lambda_\star, \mu_\star)$ as functions of $\theta$ which are well defined and differentiable according to the implicit function theorem as long as the Hessian of this Lagrangian is invertible, which we assume. Then:

$$\mathcal{H}(\theta) = \mathcal{L}(\phi_\star, \psi_\star, \lambda_\star, \mu_\star, \theta) \tag{S17}$$

$$= \frac{1}{2} ||\psi||^2 + \lambda^\top e^\psi f(\phi, \theta) - 1 + \mu^\top \nabla_\phi \mathcal{L}(\phi) \tag{S18}$$

$$= \frac{1}{2} ||\psi||^2 \tag{S19}$$

as the conditions are satisfied. We calculate the gradient:

$$d_\theta \mathcal{H}(\theta) = d_\theta \mathcal{L}(\phi_\star, \psi_\star, \lambda_\star, \mu_\star, \theta) \tag{S20}$$

$$= \partial_\theta \mathcal{L} \tag{S21}$$

$$= \lambda_\star^\top e^{\psi_\star} \partial_\theta f(\phi_\star, \theta) \tag{S22}$$

$$= -\psi_\star^\top \frac{\partial_\theta f(\phi_\star, \theta)}{f(\phi_\star, \theta)} \qquad \{\text{using condition S15}\} \tag{S23}$$

$$= \log f(\phi_\star, \theta)^\top \partial_\theta \log f(\phi_\star, \theta) \qquad \{\text{using condition S16 and log identity}\} \tag{S24}$$

$\square$

## A.4. Controlling the network dynamics

The network consists of $L$ layers with post-nonlinearity activities $\mathbf{r}_i \in \mathbb{R}^{d_i}$ for $i = 1, \dots, L$, where the input $\mathbf{r}_0$ is fixed. The controlled dynamics are

$$\tau \dot{\mathbf{r}}_i(t) = -\mathbf{r}_i(t) + e^{\psi_i(t) + \gamma_i(t)} \sigma(W_i \mathbf{r}_{i-1}(t)), \quad i = 1, \dots, L \tag{S25}$$

where $\sigma$ is an element-wise monotonically increasing nonlinearity, $W_i$ are the trainable feedforward weights, $\psi_i(t) \in \mathbb{R}^{d_i}$ is the learning signal propagated to layer $i$, and $\gamma_i(t) \in \mathbb{R}^{d_i}$ is the preservation signal.

The neuron-specific learning signal is generated by a feedback controller that continuously integrates the output error and adjusts neural activities accordingly. The optimal controller for the multiplicative case is derived in Supp. A.5 (see Supp. A.6 for comparison with the additive formulation), and convergence of the equilibrium $(\phi_\star, \psi_\star, \gamma_\star)$ is guaranteed under mild contractivity assumptions (Supp. A.7). We follow Meulemans et al. (2021)(Meulemans et al., 2021) and employ a proportional-integral (PI) controller that minimizes the discrepancy between the network's current output $\mathbf{r}_L(t)$ and a target output $\mathbf{r}_L^*$. The feedback signal and controller dynamics are

$$\psi_i(t) = Q_i \mathbf{u}(t), \quad \dot{\mathbf{u}}(t) = -(\mathbf{r}_L(t) - \mathbf{r}_L^*) - \alpha \mathbf{u}(t), \quad \mathbf{r}_L^* = \mathbf{r}_L^- - \lambda \nabla_{\mathbf{r}_L} \mathcal{L}(\mathbf{r}_L^-) \tag{S26}$$

where $\mathbf{u}(t)$ is the control signal, $\alpha$ is a leakage parameter that regularizes the controller's magnitude, and the output target $\mathbf{r}_L^*$ is defined to minimize the loss $\mathcal{L}$ either via a small step $\lambda$ along the negative loss gradient (as in weak-feedback settings (Scellier & Bengio, 2017; Meulemans et al., 2021)) or directly as the desired output (as in strong-feedback settings (Meulemans et al., 2022b; Song et al., 2024)). The feedback weights $Q_i$ can be learned via an anti-Hebbian rule when combined with noisy dynamics to estimate the Jacobian transpose $J_i^\top$ (Akrout et al., 2019; Podlaski & Machens, 2020; Meulemans et al., 2021), which we calculate and apply directly for optimal control.

Through the coupled dynamics of the network and controller, the system evolves until an equilibrium point $(\phi_\star, \psi_\star, \gamma_\star)$ is reached. Here,

$$\phi_\star \triangleq \{\mathbf{r}_{i,\star}\}_{i=1}^L, \qquad \psi_\star \triangleq \{\psi_{i,\star}\}_{i=1}^L, \qquad \gamma_\star \triangleq \{\gamma_{i,\star}\}_{i=1}^L$$

collect the equilibrium neural activities, learning signals, and preservation signals across all layers respectively, with $\psi_{i,\star} \triangleq Q_i \mathbf{u}_\star$ representing the steady-state feedback to layer $i$ and $\mathbf{u}_\star$ the steady-state controller output.

Once the joint dynamics reach equilibrium, we complete one learning step and the feedforward weights $W_i$ of each layer are updated as

$$\Delta W_i = \left(\mathbf{r}_{i,\star} - \sigma(W_i \mathbf{r}_{i-1,\star})\right) \mathbf{r}_{i-1,\star}^\top \tag{S27}$$

where $\mathbf{r}_{i-1,\star}$ and $\mathbf{r}_{i,\star}$ are the converged neural activities of the presynaptic and postsynaptic neurons, respectively. The postsynaptic residual $\mathbf{r}_{i,\star} - \sigma(W_i \mathbf{r}_{i-1,\star})$ is strictly proportional to the converged multiplicative modulation $e^{\psi_{i,\star} + \gamma_{i,\star}} - 1$, thereby translating the combined learning and preservation signals into targeted synaptic changes. At equilibrium, the output layer closely approximates the loss-minimizing target $\mathbf{r}_{L,\star} \approx \mathbf{r}_L^*$, so the task objective is embedded throughout the hidden representations via the top-down propagation of the learning signal.

We provide a pseudo-code implementation of our control framework:

---

**Algorithm 1** Equilibrium Fisher Control (EFC) — Training Loop

---

**Require:** Task sequence $\{(\mathcal{D}_1, \ldots, \mathcal{D}_T)\}$, learning rate $\eta$, preservation strength $\beta$, controller gain $k_p$, leak rate $\alpha$, convergence threshold $\epsilon$, max iterations $t_{\max}$

**Ensure:** Trained parameters $\theta$

 1: Initialize network parameters $\theta$
 2: **for** task $t = 1, \ldots, T$ **do**
 3:     **for** each mini-batch $(x, y)$ from $\mathcal{D}_t$ **do**
 4:         **— Phase 1: Feedforward pass —**
 5:         Compute feedforward activities: $\mathbf{r}_i^- \leftarrow \sigma(W_i \mathbf{r}_{i-1}^-)$ for $i = 1, \ldots, L$
 6:         Compute output target: $\mathbf{r}_L^* \leftarrow \mathbf{r}_L^- - \lambda \nabla_{\mathbf{r}_L} \mathcal{L}(\mathbf{r}_L^-, y)$
 7:
 8:         **— Phase 2: Controlled dynamics (iterate until convergence) —**
 9:         Initialize: $\mathbf{r}_i \leftarrow \mathbf{r}_i^-$, $\mathbf{u} \leftarrow \mathbf{0}$
10:         **for** $\tau = 1, \ldots, t_{\max}$ **do**
11:             Compute output error: $e \leftarrow \mathbf{r}_L - \mathbf{r}_L^*$
12:             Update controller: $\mathbf{u} \leftarrow k_p \cdot e$                     $\triangleright$ Proportional control
13:             **for** layer $i = 1, \ldots, L$ **do**
14:                 Compute learning signal: $\psi_i \leftarrow Q_i \mathbf{u}$          $\triangleright$ $Q_i = J_i^\top$ (Jacobian transpose)
15:                 Compute preservation signal: $\gamma_i \leftarrow \beta \, \mathbf{r}_{i-1}^\top F_{A,i}^D (\theta_i - \theta_{A,i}^*)$        $\triangleright$ Eq. 2
16:                 Modulate activity: $\mathbf{r}_i \leftarrow \mathbf{r}_i + \frac{\Delta t}{\tau_r} \big[ (\tanh(\psi_i + \gamma_i) + 1) \odot \sigma(W_i \mathbf{r}_{i-1}) - \mathbf{r}_i \big]$
17:             **end for**
18:             **if** $\|\Delta \mathbf{u}\| < \epsilon$ **then**
19:                 **break**                         $\triangleright$ Equilibrium reached
20:             **end if**
21:         **end for**
22:         Record equilibrium: $\mathbf{r}_{i,\star}, \psi_{i,\star}, \gamma_{i,\star}$ for all layers
23:
24:         **— Phase 3: Weight update at equilibrium —**
25:         **for** layer $i = 1, \ldots, L$ **do**
26:             $\Delta W_i \leftarrow \eta \big( \mathbf{r}_{i,\star} - \sigma(W_i \mathbf{r}_{i-1,\star}) \big) \mathbf{r}_{i-1,\star}^\top$           $\triangleright$ Hebbian-like update
27:         **end for**
28:         $\theta \leftarrow \theta + \Delta\theta$
29:     **end for**
30:
31:     **— Task boundary: store curvature —**
32:     Compute diagonal Fisher $F_t^D$ from $\mathcal{D}_t$ at current $\theta$
33:     Store $\theta_t^* \leftarrow \theta$ and accumulate $F_A^D \leftarrow F_A^D + F_t^D$
34: **end for**

---

## A.5. General Class of Controllers for Multiplicative LCP

**Theorem A.2** (General Class of Controllers for Multiplicative LCP). *Let $(\phi_\star, \psi_\star)$ be an optimal solution to the multiplicative least-control problem:*

$$\min_{\phi,\psi} \frac{1}{2}\|\psi\|^2 \quad \text{subject to} \quad e^\psi \odot f(\phi,\theta) - 1 = 0, \quad \nabla_\phi \mathcal{L}(\phi) = 0, \tag{S28}$$

*where $\odot$ denotes element-wise multiplication, $f(\phi,\theta)$ is the uncontrolled dynamics, and $\mathcal{L}(\phi)$ is the loss function. Consider the controller dynamics:*

$$\dot{\phi} = e^{Q(\phi,\theta)u} \odot f(\phi,\theta) - 1, \quad \dot{u} = -\nabla_\phi \mathcal{L}(\phi) - \alpha u, \tag{S29}$$

*with $\alpha > 0$. If at equilibrium ($\dot{\phi} = 0$, $\dot{u} = 0$), the column space of $Q(\phi_\star, \theta)$ satisfies:*

$$\text{col}[Q(\phi_\star, \theta)] = \text{row}\left[\text{diag}(f(\phi_\star, \theta))\left(\frac{\partial f}{\partial \phi}(\phi_\star, \theta)\right)^{-1}\right], \tag{S30}$$

*and $\frac{\partial f}{\partial \phi}(\phi_\star, \theta)$ is invertible, then $\psi_\star = Q(\phi_\star, \theta)u_\star$ is an optimal control for Eq. S28 in the limit $\alpha \to 0$.*

*Proof.* Let's verify that the proposed dynamics compute an optimal control $\psi_\star$. Start with the equilibrium conditions of the controller dynamics (Eq. S29). Set $\dot{\phi} = 0$ and $\dot{u} = 0$:

$$e^{Q(\phi_\star, \theta)u_\star} \odot f(\phi_\star, \theta) - 1 = 0 \implies e^{Q(\phi_\star, \theta)u_\star} \odot f(\phi_\star, \theta) = 1, \tag{S31}$$

$$-\nabla_\phi \mathcal{L}(\phi_\star) - \alpha u_\star = 0 \implies \nabla_\phi \mathcal{L}(\phi_\star) = -\alpha u_\star. \tag{S32}$$

From Eq. S32, as $\alpha \to 0$, $\nabla_\phi \mathcal{L}(\phi_\star) \to 0$ only if $u_\star$ remains finite, which we'll confirm. From Eq. S31, define $\psi_\star = Q(\phi_\star, \theta)u_\star$, so:

$$e^{\psi_\star} \odot f(\phi_\star, \theta) = 1, \tag{S33}$$

which satisfies the first constraint of Eq. S28. If $\nabla_\phi \mathcal{L}(\phi_\star) = 0$, both constraints hold, suggesting $\psi_\star$ could be optimal. We need to check optimality using the KKT conditions.

The Lagrangian for the multiplicative LCP is:

$$\mathcal{L}(\phi, \psi, \lambda, \mu) = \frac{1}{2}\|\psi\|^2 + \lambda^\top(e^\psi \odot f(\phi,\theta) - 1) + \mu^\top \nabla_\phi \mathcal{L}(\phi). \tag{S34}$$

The KKT conditions at $(\phi_\star, \psi_\star, \lambda_\star, \mu_\star)$ are:

$$\partial_\psi \mathcal{L} = \psi_\star + \lambda_\star^\top \text{diag}(e^{\psi_\star} \odot f(\phi_\star, \theta)) = 0, \tag{S35}$$

$$\partial_\phi \mathcal{L} = \left(\frac{\partial f}{\partial \phi}(\phi_\star, \theta)\right)^\top \text{diag}(e^{\psi_\star})\lambda_\star + \frac{\partial^2 \mathcal{L}}{\partial \phi^2}(\phi_\star)\mu_\star = 0, \tag{S36}$$

$$e^{\psi_\star} \odot f(\phi_\star, \theta) - 1 = 0, \tag{S37}$$

$$\nabla_\phi \mathcal{L}(\phi_\star) = 0. \tag{S38}$$

From Eq. S37, $e^{\psi_\star} \odot f(\phi_\star, \theta) = 1$, so $\text{diag}(e^{\psi_\star} \odot f(\phi_\star, \theta)) = I$ (identity matrix). Then Eq. S35 simplifies:

$$\psi_\star + \lambda_\star = 0 \implies \lambda_\star = -\psi_\star. \tag{S39}$$

Substitute into Eq. S36 with $e^{\psi_\star} = 1/f(\phi_\star, \theta)$ (component-wise, assuming $f > 0$):

$$\left(\frac{\partial f}{\partial \phi}(\phi_\star, \theta)\right)^\top \text{diag}\left(\frac{1}{f(\phi_\star, \theta)}\right)(-\psi_\star) + \frac{\partial^2 \mathcal{L}}{\partial \phi^2}(\phi_\star)\mu_\star = 0. \tag{S40}$$

Rearrange:

$$\left(\frac{\partial f}{\partial \phi}(\phi_\star, \theta)\right)^\top \text{diag}\left(\frac{1}{f(\phi_\star, \theta)}\right)\psi_\star = \frac{\partial^2 \mathcal{L}}{\partial \phi^2}(\phi_\star)\mu_\star. \tag{S41}$$

Since $\psi_\star = Q(\phi_\star, \theta)u_\star$ and $\nabla_\phi \mathcal{L}(\phi_\star) = -\alpha u_\star \to 0$, we need $\psi_\star$ to lie in the right space. Solve for $\psi_\star$:

$$\psi_\star = \text{diag}(f(\phi_\star, \theta)) \left(\frac{\partial f}{\partial \phi}(\phi_\star, \theta)\right)^{-1} \frac{\partial^2 \mathcal{L}}{\partial \phi^2}(\phi_\star)\mu_\star. \tag{S42}$$

For $\psi_\star = Qu_\star$ to hold, $Q$'s column space must span this vector. If:

$$\text{col}[Q(\phi_\star, \theta)] = \text{row}\left[\text{diag}(f(\phi_\star, \theta)) \left(\frac{\partial f}{\partial \phi}(\phi_\star, \theta)\right)^{-1}\right], \tag{S43}$$

then $\psi_\star$ can be expressed as $Qu_\star$ for some $u_\star = -\frac{1}{\alpha}\nabla_\phi \mathcal{L}(\phi_\star)$, which is finite as $\alpha \to 0$. This satisfies all KKT conditions, confirming optimality.

$\square$

## A.6. Comparison of Additive and Multiplicative LCPs

The additive and multiplicative least-control principles (LCPs) differ fundamentally in how they apply control to the system dynamics to achieve equilibrium and minimize a loss function.

- **Additive LCP**:
$$\dot{\phi} = f(\phi, \theta) + Q(\phi, \theta)u, \quad \psi_\star = -f(\phi_\star, \theta), \tag{S44}$$

$$\text{col}[Q(\phi_\star, \theta)]_{\text{additive}} = \text{row}\left[\left(\frac{\partial f}{\partial \phi}(\phi_\star, \theta)\right)^{-1}\right], \tag{S45}$$

  where the control $\psi = Qu$ is added directly to the uncontrolled dynamics $f(\phi, \theta)$.

- **Multiplicative LCP**:
$$\dot{\phi} = e^{Q(\phi, \theta)u} \odot f(\phi, \theta) - 1, \quad \psi_\star = -\log f(\phi_\star, \theta), \tag{S46}$$

$$\text{col}[Q(\phi_\star, \theta)]_{\text{multiplicative}} = \text{row}\left[\text{diag}(f(\phi_\star, \theta)) \left(\frac{\partial f}{\partial \phi}(\phi_\star, \theta)\right)^{-1}\right], \tag{S47}$$

  where the control $\psi = Qu$ scales $f(\phi, \theta)$ component-wise via the exponential $e^\psi$.

The key difference lies in the control mechanism. In the additive LCP, $\psi$ acts as a linear counterforce, directly offsetting $f(\phi, \theta)$ to achieve $\dot{\phi} = 0$, and $Q$ aligns with the inverse Jacobian to map the control input $u$ efficiently. In contrast, the multiplicative LCP modulates $f(\phi, \theta)$ exponentially, requiring $\psi$ to adjust each component's magnitude to satisfy $e^{\psi_\star} \odot f(\phi_\star, \theta) = 1$. The additional $\text{diag}(f(\phi_\star, \theta))$ term in the multiplicative column space condition accounts for this scaling: larger components of $f(\phi_\star, \theta)$ need a proportionally stronger (often more negative) $\psi$ to reduce their effect, which $Q$ must facilitate. Thus, while the additive approach pushes the dynamics uniformly, the multiplicative approach tunes them selectively, making the dynamics sensitive to the size of $f(\phi, \theta)$.

## A.7. Convergence and equilibrium points

**Theorem A.3** (Convergence and equilibrium points). *Let $(\phi_\star, \psi_\star)$ be an optimal control for the multiplicative least-control problem of Eq. S28, and let $(\lambda_\star, \mu_\star)$ be the Lagrange multipliers for which the KKT conditions are satisfied. For $\beta > 0$, the system converges to a local equilibrium point characterized by:*

$$\log f(\phi_\star, \theta) = -\beta F_A(\hat{\theta} - \theta_A) \tag{S48}$$

*This equilibrium balances the learning objective with the Fisher preservation term.*

*Proof.* With the constraint $e^{\psi + \gamma} f(\phi, \theta) = 1$ we solve $\psi = -\gamma - \log f(\phi, \theta)$, whereafter the Lyapunov function becomes:

$$V(\theta) = \frac{1}{2}||\psi_\star(\theta)||^2 = \frac{1}{2}\left(-\gamma - \log f(\phi_\star, \theta)\right)^2 \tag{S49}$$

together with the result from Theorem A.1, namely $\dot{\theta} = -d_\theta H(\theta) = -\log f(\phi_\star, \theta)^\top \partial_\theta \log f(\phi_\star, \theta)$, we find $\dot{V}$:

$$\dot{V}(\theta) = \frac{d}{dt}\left[\frac{1}{2}\left(-\gamma - \log f(\phi_\star, \theta)\right)^2\right] \tag{S50}$$

$$= (-\gamma - \log f(\phi_\star, \theta))(-\partial_\theta \log f(\phi_\star, \theta))^\top \dot{\theta} \tag{S51}$$

$$= -(\gamma + \log f(\phi_\star, \theta))(-\log f(\phi_\star, \theta))^\top (-\log f(\phi_\star, \theta)^\top \partial_\theta \log f(\phi_\star, \theta)) \tag{S52}$$

$$= -(\gamma \log f(\phi_\star, \theta) + (\log f(\phi_\star, \theta))^2)||\partial_\theta \log f(\phi_\star, \theta)||^2 \tag{S53}$$

now we can describe the conditions for $\dot{V} = 0$, either (1) $(\gamma \log f(\phi_\star, \theta) + (\log f(\phi_\star, \theta))^2) = 0$, or (2) $||\partial_\theta \log f(\phi_\star, \theta)||^2 = 0$. For (1):

$$(\gamma \log f(\phi_\star, \theta) + (\log f(\phi_\star, \theta))^2) = 0 \implies f(\phi_\star, \theta) = 1 \ \lor \ f(\phi_\star, \theta) = e^{-\gamma} \tag{S54}$$

where the conditions imply that the learning has converged or that the learning is balanced with the Fisher preservation term, respectively. For (2) simply $\partial_\theta \log f(\phi_\star, \theta) = 0$, meaning no further updates to $\theta$. □

**Corollary A.4** (Special case: $\beta = 0$). *When the Fisher preservation term is absent ($\beta = 0$), the system reduces to pure learning without any continual learning constraints. In this case, the equilibrium point has a simple form that only reflects the learning objective:*

$$\log f(\phi_\star, \theta) = 0 \tag{S55}$$

*Proof.* When $\beta = 0$, we can find the equilibrium point in two equivalent ways. First, from the Lyapunov analysis:

$$(\gamma \log f(\phi_\star, \theta) + (\log f(\phi_\star, \theta))^2) = 0 \tag{S56}$$
$$\log f(\phi_\star, \theta) = 0 \tag{S57}$$

Alternatively, we can derive the same result from the general equilibrium condition. When $\beta = 0$, the Fisher preservation term vanishes:

$$\log f(\phi_\star, \theta) = -\beta F_A(\hat{\theta} - \theta_A) \tag{S58}$$
$$= 0 \tag{S59}$$

□

**Corollary A.5** (Special case: $\beta \neq 0$). *When the Fisher preservation term is active ($\beta \neq 0$), the learned state achieved in Corollary A.4 becomes impossible to reach.*

*Proof.* We proceed by contradiction. Assume that perfect learning ($\log f(\phi_\star, \theta) = 0$) can be achieved when $\beta \neq 0$. This leads to three mutually exclusive possibilities, each of which yields a contradiction:

(1) $\beta = 0$: This directly contradicts our assumption that $\beta \neq 0$ ($\perp$)

(2) $\hat{\theta} - \theta_A = 0$: This would mean the parameters haven't changed from their values on Task A, which makes the equilibrium point non-unique since we could achieve zero error without parameter movement ($\perp$)

(3) $F_A = 0$: This would imply that the Fisher Information Matrix is zero, meaning the parameters have no effect on the Task A performance, contradicting our assumption that the parameters are meaningful for learning ($\perp$) □

## B. Steady-state of the network

In this section, we aim to explore the steady-state of the EFC framework. Using the steady-state, we can proof the continual-natural gradient property of the learning dynamics in the subsequent section, thereafter allowing us to make theoretical claims regarding the forgetting comparison.

### B.1. Steady-state approximation of multiplicative-LCP with preservation

The crux of our learning framework is that we only update the weights at equilibrium, $(\psi_\star, \phi_\star)$. In order to obtain this equilibrium point we can run the dynamics until convergence where no meaningful changes occur, or alternatively we can approximate the equilibrium by approximating the steady-state solution of the network dynamics directly. Theorem B.1 describes the steady-state first-order Taylor approximation of the network dynamics defined by **Eq. S26**. Intuitively, the individual neurons receive both a learning and a preservation signal: $Q\mathbf{u}_\star + \gamma$. The control signal $\mathbf{u}_\star$ is responsible for driving the neural activities to solve the objective, yet is hindered by the cumulative effect of the preservation signals $\gamma_{\text{eff}}$, and therefore integrates the balance between the learning and preservation: $\delta_L^- - \gamma_{\text{eff}}$. The direction where $\mathbf{u}_\star$ pushes toward is determined by the cumulative effect of the individual neurons on the output neurons, $J_{\text{eff}}$. Namely, this effect is dynamically inverted – constrained by the column space of $Q$ (Meulemans et al., 2021; Podlaski & Machens, 2020) – to obtain the appropriate effect of the output neurons on the individual neurons. Practically, this results in steady-state neural activities $\mathbf{r}_\star$ that are multiplied around the feedforward activity $\mathbf{r}^-$, proportional to the preservation signal and its interaction with learning: $\text{diag}(\mathbf{r}^-)(Q\mathbf{u}_\star + \gamma)$.

The steady-state proof follows a similar structure as in (Meulemans et al., 2021). We derive the steady-state control for a neural network governed by multiplicative-LCP dynamics with a preservation term, directly from the dynamical equations.

- Layer dynamics for $i \in \{1, \ldots, L\}$:

$$\tau_r \dot{\mathbf{r}}_i(t) = -\mathbf{r}_i(t) + e^{Q_i \mathbf{u}(t) + \gamma_i} \cdot \phi\left(W_i \mathbf{r}_{i-1}(t)\right) \tag{S60}$$

where $\mathbf{r}_i(t)$ is the state of layer $i$, $\mathbf{r}_0(t)$ is the fixed input, $Q_i$ is the control matrix for layer $i$, $\mathbf{u}(t)$ is the control input, $\gamma_i$ is a preservation term, $\phi$ is the activation function, and $W_i$ is the weight matrix from layer $i-1$ to $i$.

- Control dynamics:

$$\dot{\mathbf{u}} = -(\mathbf{r}_L - \mathbf{r}_L^*) - \alpha \mathbf{u} \tag{S61}$$

where $\mathbf{r}_L^*$ is the target output for the final layer, and $\alpha > 0$ is a damping coefficient.

Our objective is to determine the steady-state control $\mathbf{u}_\star$ by linearizing around the feedforward state, incorporating the initial output error and preservation terms across all layers.

**Theorem B.1** (Steady-state approximation for multiplicative LCP). *Under a first-order linear approximation around the feedforward state $\mathbf{r}^-$, the steady-state solutions are:*

$$\mathbf{u}_\star = (J_{\text{eff}} + \alpha I)^{-1}(\delta_L^- - \gamma_{\text{eff}}), \tag{S62}$$

$$\mathbf{r}_\star = \mathbf{r}^- + (I - J)^{-1} \text{diag}(\mathbf{r}^-)(Q\mathbf{u}_\star + \gamma), \tag{S63}$$

*where $\mathbf{r}^- = [\mathbf{r}_1^{-\top}, \mathbf{r}_2^{-\top}, \ldots, \mathbf{r}_L^{-\top}]^\top$ satisfies $\mathbf{r}_i^- = \phi(W_i \mathbf{r}_{i-1}^-)$ with $\mathbf{r}_0^- = \mathbf{r}_0$, $\delta_L^- = \mathbf{r}_L^* - \mathbf{r}_L^-$ is the initial output error, $J_{\text{eff}} = \sum_{i=1}^{L} \left(\prod_{k=i+1}^{L} J_{k,k-1}\right)(Q_i \odot \mathbf{r}_i^-)$ is the effective Jacobian from control to output, $\gamma_{\text{eff}} = \sum_{i=1}^{L} \left(\prod_{k=i+1}^{L} J_{k,k-1}\right)(\gamma_i \odot \mathbf{r}_i^-)$ is the effective preservation term, $J_{i,i-1} = \phi'(W_i \mathbf{r}_{i-1}^-) \odot W_i$ is the local Jacobian, $J$ is the block lower-triangular matrix with $J_{i,i-1}$ on the subdiagonal, $Q = [Q_1^\top, Q_2^\top, \ldots, Q_L^\top]^\top$ stacks the feedback matrices, $\gamma = [\gamma_1^\top, \gamma_2^\top, \ldots, \gamma_L^\top]^\top$ stacks the preservation terms, and $\text{diag}(\mathbf{r}^-)$ is the block-diagonal matrix of feedforward states.*

*Proof.* At steady state, meaning $\dot{\mathbf{r}}_i = 0$ and $\dot{\mathbf{u}} = 0$, yielding for each layer $i$:

$$\mathbf{r}_{i,\star} = e^{Q_i \mathbf{u}_\star + \gamma_i} \cdot \phi\left(W_i \mathbf{r}_{i-1,\star}\right) \tag{S64}$$

$$\mathbf{r}_{L,\star} = \mathbf{r}_L^* - \alpha \mathbf{u}_\star \tag{S65}$$

We now want to perturb the feedforward state by a small $\mathbf{u}_\star$ and $\gamma_i$. The feedforward state, when $\mathbf{u} = 0$ and $\gamma_i = 0$, is defined by:

$$\mathbf{r}_i^- = \phi\left(W_i \mathbf{r}_{i-1}^-\right), \quad \mathbf{r}_0^- = \mathbf{r}_0, \quad \delta_L^- = \mathbf{r}_L^* - \mathbf{r}_L^- \tag{S66}$$

with $\delta_L$ being the initial output error. Now perturb and linearize around $\mathbf{r}_i^-$:

$$\mathbf{r}_i^- + \Delta \mathbf{r}_i = e^{Q_i \mathbf{u}_\star + \gamma_i} \cdot \phi\left(W_i(\mathbf{r}_{i-1}^- + \Delta \mathbf{r}_{i-1})\right) \tag{S67}$$

$$\tag{S68}$$

where we can approximate both parts,

$$e^{Q_i \mathbf{u}_\star + \gamma_i} \approx 1 + Q_i \mathbf{u}_\star + \gamma_i \tag{S69}$$

$$\phi\left(W_i(\mathbf{r}_{i-1}^- + \Delta \mathbf{r}_{i-1})\right) = \phi\left(W_i \mathbf{r}_{i-1}^- + W_i \Delta \mathbf{r}_{i-1}\right) \tag{S70}$$

$$\approx \mathbf{r}_i^- + \phi'\left(W_i \mathbf{r}_{i-1}^-\right) \odot W_i \Delta \mathbf{r}_{i-1} \tag{S71}$$

$$\tag{S72}$$

now substituting,

$$\mathbf{r}_i^- + \Delta \mathbf{r}_i \approx (1 + Q_i \mathbf{u}_\star + \gamma_i)\left(\mathbf{r}_i^- + \phi'\left(W_i \mathbf{r}_{i-1}^-\right) \odot W_i \Delta \mathbf{r}_{i-1}\right) \tag{S73}$$

$$\Delta \mathbf{r}_i \approx (Q_i \mathbf{u}_\star + \gamma_i)\mathbf{r}_i^- + (1 + Q_i \mathbf{u}_\star + \gamma_i)\left(\phi'\left(W_i \mathbf{r}_{i-1}^-\right) \odot W_i \Delta \mathbf{r}_{i-1}\right) \tag{S74}$$

$$\tag{S75}$$

now expanding and noticing that the cross-term $(Q_i \mathbf{u}_\star + \gamma_i)\left(\phi'\left(W_i \mathbf{r}_{i-1}^-\right) \odot W_i \Delta \mathbf{r}_{i-1}\right)$ is negligible by our assumptions we get,

$$\Delta \mathbf{r}_i \approx (Q_i \mathbf{u}_\star + \gamma_i) \odot \mathbf{r}_i^- + \phi'\left(W_i \mathbf{r}_{i-1}^-\right) \odot W_i \Delta \mathbf{r}_{i-1} \tag{S76}$$

Define the local Jacobian as $J_{i,i-1} = \phi'\left(W_i \mathbf{r}_{i-1}^-\right) \odot W_i$,

$$\Delta \mathbf{r}_i \approx J_{i,i-1} \Delta \mathbf{r}_{i-1} + (Q_i \mathbf{u}_\star + \gamma_i) \odot \mathbf{r}_i^- \tag{S77}$$

$$\Delta \mathbf{r}_i - J_{i,i-1} \Delta \mathbf{r}_{i-1} \approx (Q_i \mathbf{u}_\star + \gamma_i) \odot \mathbf{r}_i^- \tag{S78}$$

We can now make a general form by taking $\Delta \mathbf{r}_0 = 0$ and stacking the perturbations $\Delta \mathbf{r} = [\Delta \mathbf{r}_1^\top, \Delta \mathbf{r}_2^\top, \ldots, \Delta \mathbf{r}_L^\top]^\top$ across all layers:

$$(I - J)\Delta \mathbf{r} = \text{diag}(\mathbf{r}^-)(Q\mathbf{u}_\star + \gamma) \tag{S79}$$

$$\Delta \mathbf{r} = (I - J)^{-1}\text{diag}(\mathbf{r}^-)(Q\mathbf{u}_\star + \gamma) \tag{S80}$$

therefore,

$$\mathbf{r}_\star = \mathbf{r}^- + (I - J)^{-1}\text{diag}(\mathbf{r}^-)(Q\mathbf{u}_\star + \gamma) \tag{S81}$$

Now we need to find the steady-state approximation for the control signal. We will do so by equating two definitions of the controller's effect on the output. First,

$$\Delta \mathbf{r}_L = \mathbf{r}_{L,\star} - \mathbf{r}_L^- = \mathbf{r}_L^* - \alpha \mathbf{u}_\star - \mathbf{r}_L^- = \delta_L^- - \alpha \mathbf{u}_\star \tag{S82}$$

Second, from the propagated effect of $\mathbf{u}_\star$ across the network. Since we have a layered and thereby recursive network structure, we obtain the effective cumulative sensitivity of $\mathbf{r}_L$ to $\mathbf{u}_\star$ as well as the cumulative effect of $\gamma$'s effect on the output, giving:

$$J_{\text{eff}} = \sum_{i=1}^{L} \left(\prod_{k=i+1}^{L} J_{k,k-1}\right)(Q_i \odot \mathbf{r}_i^-) \tag{S83}$$

$$\gamma_{\text{eff}} = \sum_{i=1}^{L} \left(\prod_{k=i+1}^{L} J_{k,k-1}\right)(\gamma_i \odot \mathbf{r}_i^-) \tag{S84}$$

$$\Delta \mathbf{r}_L \approx J_{\text{eff}} \mathbf{u}_\star + \gamma_{\text{eff}} \tag{S85}$$

with identity matrices for $i = L$.

We can now equate the two expressions for $\Delta \mathbf{r}_L$ and solve:

$$\delta_L^- - \alpha \mathbf{u}_\star = J_{\text{eff}} \mathbf{u}_\star + \gamma_{\text{eff}} \tag{S86}$$

$$(J_{\text{eff}} + \alpha I) \mathbf{u}_\star = \delta_L^- - \gamma_{\text{eff}} \tag{S87}$$

$$\mathbf{u}_\star = (J_{\text{eff}} + \alpha I)^{-1} (\delta_L^- - \gamma_{\text{eff}}) \tag{S88}$$

$\square$

# C. Continual-natural gradient

Now that we have derived the steady-state approximation of our dynamical system, we can combine this result with the equilibrium states of Supp. A.2 to obtain the continual-natural gradient property which will be used for the theoretical loss and forgetting bounds in the following sections.

## C.1. Continual-natural gradient property of the learning dynamics

What we call the *continual-natural gradient* property of the learning dynamics is intuitively the observation that during the learning of Task B, the parameters are constraint by the local geometry learning by Task A. In the following theorem, we will explore that the neuron-specific preservation signals $\gamma_i$ interact through the network dynamics, and while the individual $\gamma$'s are only derived from the diagonal of the Fisher Information Metric, the resulting dynamics through both feedforward and feedback interactions through the controller, are sufficient to approximate the full Fisher matrix.

**Theorem C.1** (Continual-natural gradient property of EFC learning dynamics). *Define the preservation signal for layer $i$ as $\gamma_i = -\beta F_{A,i}^D(\theta - \theta_A^*)$, where $\beta > 0$ is a scalar weighting factor, and the effective preservation derived from the steady-state of Theorem B.1 signal as:*

$$\gamma_{\textit{eff}} = \sum_{i=1}^{L} J_i(\gamma_i \odot \mathbf{r}_i^-), \tag{S89}$$

*where $J_i = \prod_{k=i+1}^{L} J_{k,k-1}$ is the Jacobian of the network output $\mathbf{r}_L$ with respect to the pre-activation output $\mathbf{r}_i^- = W_i \mathbf{r}_{i-1}$ of layer $i$, $J_{k,k-1} = \phi'(W_k \mathbf{r}_{k-1}) \odot W_k$, $\phi'$ is the derivative of the activation function, $W_k$ is the weight matrix, and $\odot$ denotes element-wise multiplication. Let the feedback matrices be $Q_i = J_i^T$. Assuming small learning rate $\eta$, small regularization parameter $\alpha$ in the control signal, and linearization around $\theta_A^*$, the weight update in the EFC framework satisfies:*

$$\Delta\theta \approx -\eta \tilde{F}_A^{-1} \nabla_\theta \mathcal{L}_B, \tag{S90}$$

*where $\tilde{F}_A = J^\top J_{\textit{eff}}^{-1} \sum_{i=1}^{L} J_i(F_{A,i}^D \odot \mathbf{r}_i^-)$.*

*Proof.* First we recap the setup, in the EFC framework the dynamics are:

$$\tau_i \dot{\mathbf{r}}_i(t) = -\mathbf{r}_i(t) + e^{Q_i \mathbf{u}(t) + \gamma_i} \phi(W_i \mathbf{r}_{i-1}(t)), \tag{S91}$$

$$\dot{\mathbf{u}}(t) = -(\mathbf{r}_L(t) - \mathbf{r}_L^*) - \alpha \mathbf{u}(t), \tag{S92}$$

with $\mathbf{u}(t)$ as the control signal, $\mathbf{r}_L^*$ as the target output for Task B, and $\alpha > 0$ as a damping coefficient. At equilibrium $(\dot{\mathbf{r}}_i = 0, \dot{\mathbf{u}} = 0)$:

$$\mathbf{r}_{i,*} = e^{Q_i \mathbf{u}_* + \gamma_i} \phi(W_i \mathbf{r}_{i-1,*}) \tag{S93}$$

$$\mathbf{r}_{L,*} = \mathbf{r}_L^* - \alpha \mathbf{u}_* \tag{S94}$$

$$\psi_* = Q \mathbf{u}_* \tag{S95}$$

$$\Delta\theta_{\text{EFC}} = -\eta \nabla_\theta \mathcal{H}(\theta) = -\eta \frac{\partial \psi_*}{\partial \theta} \psi_* \tag{S96}$$

Set $Q = \left[J_1^T, J_2^T, \ldots, J_L^T\right]^\top$, where:

$$J_i = \prod_{k=i+1}^{L} J_{k,k-1}, \quad J_{k,k-1} = \phi'(W_k \mathbf{r}_{k-1}) \odot W_k, \tag{S97}$$

The effective preservation signal is (assume $\beta = 1$ for simplicity):

$$\gamma_{\text{eff}} = \sum_{i=1}^{L} J_i(\gamma_i \odot \mathbf{r}_i^-), \quad \gamma_i = -F_{A,i}^D(\theta - \theta_A^*) \implies \gamma_{\text{eff}} = \sum_{i=1}^{L} J_i(-F_{A,i}^D(\theta - \theta_A^*) \odot \mathbf{r}_i^-) \tag{S98}$$

The steady-state control signal is:

$$\mathbf{u}_* = (J_{\text{eff}} + \alpha I)^{-1}(\delta_L^- - \gamma_{\text{eff}}), \tag{S99}$$

where $\delta_L^- = \mathbf{r}_L^* - \mathbf{r}_L^- \approx -J\nabla_\theta\mathcal{L}_B$, and $J$ is the full Jacobian from $\theta$ to $\mathbf{r}_L$. For small $\alpha$:

$$\mathbf{u}_* \approx J_{\text{eff}}^{-1}(-J\nabla_\theta\mathcal{L}_B - \gamma_{\text{eff}}). \tag{S100}$$

Thus with $Q = J^\top$:

$$\psi_* = Q\mathbf{u}_* = J^\top J_{\text{eff}}^{-1}(-J\nabla_\theta\mathcal{L}_B - \gamma_{\text{eff}}). \tag{S101}$$

EFC minimizes $\mathcal{H}(\theta)$ at equilibrium:

$$Q J_{\text{eff}}^{-1}(-J\nabla_\theta\mathcal{L}_B - \gamma_{\text{eff}}) = 0 \tag{S102}$$

$$-J\nabla_\theta\mathcal{L}_B = \gamma_{\text{eff}} \tag{S103}$$

$$= \sum_{i=1}^L J_i(-F_{A,i}^D(\theta - \theta_A^*) \odot \mathbf{r}_i^-) \tag{S104}$$

Since $\Delta\theta = \theta - \theta_A^*$:

$$\nabla_\theta\mathcal{L}_B = J^\top J_{\text{eff}}^{-1} \sum_{i=1}^L J_i(F_{A,i}^D(\theta - \theta_A^*) \odot \mathbf{r}_i^-) \tag{S105}$$

we define

$$\tilde{F}_A = J^\top J_{\text{eff}}^{-1} \sum_{i=1}^L J_i(F_{A,i}^D \odot \mathbf{r}_i^-) \tag{S106}$$

Now we reach our desired statement:

$$\Delta\theta \approx -\tilde{F}_A^{-1}\nabla_\theta\mathcal{L}_B \tag{S107}$$

However, we are still burdened with the question whether our defined $\tilde{F}_A$ is an apt approximation of the true $F_A$. Now, to confirm $\tilde{F}_A$ approximates the true Fisher $F_A = J_A^\top J_A$, where $J_A = \frac{\partial \mathbf{r}_L}{\partial \theta}$ is evaluated under Task A's distribution, note that $F_A$ captures the full curvature of Task A's loss in parameter space, including off-diagonal interactions. In EFC, $\tilde{F}_A = J^\top J_{\text{eff}}^{-1} \sum_{i=1}^L J_i(F_{A,i}^D \odot \mathbf{r}_i^-)$ leverages:

- $F_{A,i}^D$, the diagonal Fisher per layer, encoding parameter importance,

- $\text{diag}(\mathbf{r}_i^-)$, modulating preservation strength dynamically,

- $J_i$, projecting layer-wise effects to the output, and $J^\top J_{\text{eff}}^{-1}$, embedding network dynamics via $J_{\text{eff}} = \sum_{i=1}^L J_i\text{diag}(\mathbf{r}_i^-)J_i^\top$.

Unlike a static diagonal approximation, $J_{\text{eff}}^{-1}$ introduces effective inter-layer dependencies, approximating $F_A$'s off-diagonal terms through the control signal's modulation across layers. Thus, $\tilde{F}_A$ dynamically captures both diagonal and off-diagonal curvature of $F_A$, justifying the continual-natural gradient property. $\qquad\square$

We argue $\tilde{F}_A$ adequately reflects the structure of the full $F_A$ as follows. First, the preservation signals $\gamma_i \propto -F_{A,i}^D$ embed Task A's diagonal Fisher into Task B's dynamics, forward through $\sum_{i=1} J_i(F_{A,i}^D \odot \mathbf{r}_i^-) \propto \gamma_{\text{eff}}$ and backward through $J_{\text{eff}}^{-1}$ within the control signal. Now these parameter interactions are not yet sufficient to ensure the structure of Task A is accounted for in the weight update. Thus, we observe that the weight update at equilibrium is of minimum norm (Meulemans et al., 2020). A minimum norm can effectively be viewed as a budget which the controller has to spend wisely. Here, the embedding of Task A's interactions into the dynamics of Task B results in a growing cost for interaction effects that do not align with $\nabla_\theta\mathcal{L}_B$, thereby ensuring parameters preferentially update orthogonal to Task A's full curvature. Therefore, $\tilde{F}_A$ approximates $F_A$ well at equilibrium with least control.

# D. Class-Incremental Convergence

This section provides derivations supporting Section 3.2.1 and Section 3.2.2. Section D.1 derives the decomposition $\nabla_\theta \mathcal{L}_B(x_B) = G_B(x_B) + G_{A \leftarrow B}(x_B)$ for standard losses. Section D.3 places $G_{A \leftarrow B}(x_B)$ in the Task A-sensitive subspace. Section D.2 formalizes the limitation of parameter-only regularization. Section D.5 bounds the change in the joint class-incremental objective under EFC.

## D.1. Gradient decomposition for shared output heads

Consider a neural network $f_\theta : \mathcal{X} \to \mathbb{R}^{|\mathcal{C}|}$ with classes $\mathcal{C} = \mathcal{C}_A \cup \mathcal{C}_B$ and logits $z = f_\theta(x)$. For a Task B sample $(x_B, y_B)$ with $y_B \in \mathcal{C}_B$, the gradient $\nabla_\theta \mathcal{L}_B(x_B, y_B)$ decomposes into a Task B term and a shared-head interference term.

The cross-entropy loss is

$$\mathcal{L}(x, y) = -\log p_y, \qquad p_c = \frac{e^{z_c}}{\sum_{c' \in \mathcal{C}} e^{z_{c'}}}. \tag{S108}$$

The parameter gradient is

$$\nabla_\theta \mathcal{L}(x, y) = \sum_{c \in \mathcal{C}} (p_c - \mathbf{1}_{c=y}) \nabla_\theta z_c(x). \tag{S109}$$

For $y_B \in \mathcal{C}_B$,

$$\nabla_\theta \mathcal{L}(x_B, y_B) = \underbrace{\sum_{c \in \mathcal{C}_B} (p_c - \mathbf{1}_{c=y_B}) \nabla_\theta z_c(x_B)}_{G_B(x_B)} + \underbrace{\sum_{c \in \mathcal{C}_A} p_c \nabla_\theta z_c(x_B)}_{G_{A \leftarrow B}(x_B)}. \tag{S110}$$

For mean-squares error with, one-hot targets $\mathbf{y}$,

$$\mathcal{L}(x, y) = \frac{1}{2} \sum_{c \in \mathcal{C}} (z_c - \mathbf{y}_c)^2, \qquad \nabla_\theta \mathcal{L}(x, y) = \sum_{c \in \mathcal{C}} (z_c - \mathbf{y}_c) \nabla_\theta z_c(x). \tag{S111}$$

For $y_B \in \mathcal{C}_B$,

$$\nabla_\theta \mathcal{L}(x_B, y_B) = \underbrace{\sum_{c \in \mathcal{C}_B} (z_c - \mathbf{1}_{c=y_B}) \nabla_\theta z_c(x_B)}_{G_B(x_B)} + \underbrace{\sum_{c \in \mathcal{C}_A} z_c \nabla_\theta z_c(x_B)}_{G_{A \leftarrow B}(x_B)}. \tag{S112}$$

Equations (S110) and (S112) establish

$$\nabla_\theta \mathcal{L}_B(x_B) = G_B(x_B) + G_{A \leftarrow B}(x_B), \tag{S113}$$

with $G_{A \leftarrow B}(x_B) = \sum_{c \in \mathcal{C}_A} w_c(x_B) \nabla_\theta z_c(x_B)$ for weights $w_c(x_B) \geq 0$.

## D.2. Parameter-based regularization cannot cancel the interference term

This section formalizes a limitation for methods that modify backpropagation only by adding a gradient $\nabla_\theta \mathcal{R}(\theta)$, where $\mathcal{R}$ depends only on $\theta$.

**Lemma D.1** (Sample-dependent interference cannot be cancelled). *Let $\mathcal{R} : \mathbb{R}^{|\theta|} \to \mathbb{R}$ be differentiable. Assume existence of $x_1, x_2 \in \mathcal{D}_B$ such that $G_{A \leftarrow B}(x_1) \neq G_{A \leftarrow B}(x_2)$. No parameter vector $\theta$ satisfies $\nabla_\theta \mathcal{R}(\theta) = -G_{A \leftarrow B}(x)$ for all $x \in \mathcal{D}_B$.*

*Proof.* The vector $\nabla_\theta \mathcal{R}(\theta)$ is uniquely determined by $\theta$. Two distinct equalities $\nabla_\theta \mathcal{R}(\theta) = -G_{A \leftarrow B}(x_1)$ and $\nabla_\theta \mathcal{R}(\theta) = -G_{A \leftarrow B}(x_2)$ cannot both hold when $G_{A \leftarrow B}(x_1) \neq G_{A \leftarrow B}(x_2)$. $\qquad \square$

Lemma D.1 establishes that cancellation of $G_{A \leftarrow B}(x_B)$ on a per-sample basis is unavailable to any penalty term depending only on $\theta$.

A second statement addresses subspace-selective attenuation at the level of per-sample updates. Parameter-based regularization produces an additive update of the form $-\eta(\nabla_\theta \mathcal{L}_B(x_B) + \nabla_\theta \mathcal{R}(\theta))$. A preconditioned update has the form $-\eta P \nabla_\theta \mathcal{L}_B(x_B)$ with a matrix $P$ multiplying the data gradient. Additive penalties do not multiply $\nabla_\theta \mathcal{L}_B(x_B)$ and therefore do not implement per-sample subspace filtering of $G_{A \leftarrow B}(x_B)$.

**Corollary D.2** (Stationary condition of regularized Task B training). *Any stationary point of $\min_\theta \mathbb{E}_{x_B}[\mathcal{L}_B(x_B; \theta)] + \mathcal{R}(\theta)$ satisfies*

$$\mathbb{E}_{x_B}[G_B(x_B) + G_{A \leftarrow B}(x_B)] + \nabla_\theta \mathcal{R}(\theta) = 0. \tag{S114}$$

As a remark, for a second-order optimization of $\mathbb{E}[\mathcal{L}_B] + \mathcal{R}$, this uses local curvature of $\mathcal{L}_B$ and $\mathcal{R}$ through $(\nabla^2 \mathcal{L}_B + \nabla^2 \mathcal{R})^{-1}$. The resulting update depends on Task B curvature and does not isolate $G_{A \leftarrow B}(x_B)$ from the remaining components of $\nabla_\theta \mathcal{L}_B(x_B)$. The decomposition $G_B(x_B) + G_{A \leftarrow B}(x_B)$ is not identifiable from the sum alone.

### D.3. The interference term lies in the Task A-sensitive subspace

Let $\mathcal{V}_A \subseteq \mathbb{R}^{|\theta|}$ denote the Task A-sensitive subspace. Define

$$F_A = \mathbb{E}_{x \sim \mathcal{D}_A}[\nabla_\theta \mathcal{L}_A(x) \nabla_\theta \mathcal{L}_A(x)^\top], \qquad \mathcal{V}_A = \text{col}(F_A). \tag{S115}$$

**Lemma D.3** (Alignment of interference gradient). *For any $x_B \in \mathcal{D}_B$, the interference term satisfies $G_{A \leftarrow B}(x_B) \in \mathcal{V}_A$.*

*Proof.* Section D.1 gives $G_{A \leftarrow B}(x_B) = \sum_{c \in \mathcal{C}_A} w_c(x_B) \nabla_\theta z_c(x_B)$ with $w_c(x_B) \geq 0$. For Task A samples, the gradients $\nabla_\theta \mathcal{L}_A(x)$ are linear combinations of $\{\nabla_\theta z_c(x) : c \in \mathcal{C}_A\}$. The column space $\mathcal{V}_A = \text{col}(F_A)$ is the span of Task A gradients, hence contains directions affecting Task A logits. The term $G_{A \leftarrow B}(x_B)$ is a linear combination of Task A logit gradients and belongs to the same span. $\square$

### D.4. Spectral attenuation under EFC

Let $\mathcal{V}_A = \text{col}(F_A)$ and let $P_A$ denote the orthogonal projector onto $\mathcal{V}_A$. Let $P_A^\perp = I - P_A$. Decompose

$$G_B(x_B) = G_B^\|(x_B) + G_B^\perp(x_B), \qquad G_B^\|(x_B) = P_A G_B(x_B), \qquad G_B^\perp(x_B) = P_A^\perp G_B(x_B). \tag{S116}$$

Lemma D.3 gives $G_{A \leftarrow B}(x_B) \in \mathcal{V}_A$.

Let $\tilde{F}_A$ denote the EFC curvature matrix from Theorem 3.1. Let $\tilde{\lambda}_{\min}$ denote the minimum eigenvalue of $\tilde{F}_A$ restricted to $\mathcal{V}_A$.

**Lemma D.4** (Attenuation on $\mathcal{V}_A$). *For any $v \in \mathcal{V}_A$,*

$$\|\tilde{F}_A^{-1} v\| \leq \tilde{\lambda}_{\min}^{-1} \|v\|. \tag{S117}$$

*Proof.* Restrict $\tilde{F}_A$ to $\mathcal{V}_A$ and diagonalize the restriction. All eigenvalues on $\mathcal{V}_A$ are at least $\tilde{\lambda}_{\min}$ by definition. The operator norm of the restricted inverse is at most $\tilde{\lambda}_{\min}^{-1}$. $\square$

**Lemma D.5** (EFC update decomposition). *For any $x_B$, the EFC update satisfies*

$$\Delta\theta_{\text{EFC}}(x_B) = -\eta \, \tilde{F}_A^{-1} G_B^\perp(x_B) + \varepsilon(x_B), \qquad \|\varepsilon(x_B)\| \leq \eta \, \tilde{\lambda}_{\min}^{-1} \|G_B^\|(x_B) + G_{A \leftarrow B}(x_B)\|. \tag{S118}$$

*Proof.* Start from $\Delta\theta_{\text{EFC}}(x_B) = -\eta \, \tilde{F}_A^{-1}(G_B(x_B) + G_{A \leftarrow B}(x_B))$. Decompose $G_B(x_B) = G_B^\|(x_B) + G_B^\perp(x_B)$. Lemma D.3 gives $G_{A \leftarrow B}(x_B) \in \mathcal{V}_A$ and $G_B^\|(x_B) \in \mathcal{V}_A$ by definition. Lemma D.4 bounds $\tilde{F}_A^{-1}$ on $\mathcal{V}_A$ and gives the stated estimate. $\square$

### D.5. Convergence on the joint class-incremental objective

Define the joint objective

$$\mathcal{L}_{A \cup B}(\theta) = \mathbb{E}_{x_A \sim \mathcal{D}_A}[\mathcal{L}_A^{\mathcal{C}_A \cup \mathcal{C}_B}(x_A; \theta)] + \mathbb{E}_{x_B \sim \mathcal{D}_B}[\mathcal{L}_B^{\mathcal{C}_A \cup \mathcal{C}_B}(x_B; \theta)]. \tag{S119}$$

**Theorem D.6** (One-step descent under EFC). *Assume differentiability of $\mathcal{L}_{A \cup B}$ at $\theta$ and bounded second derivatives in a neighborhood of $\theta$. Let $\Delta\theta_{\text{EFC}}(x_B)$ denote the EFC update for sample $x_B$. Let $\tilde{\lambda}_{\min}$ denote the minimum eigenvalue of $\tilde{F}_A$ restricted to $\mathcal{V}_A$. Then the expected change satisfies*

$$\mathbb{E}_{x_B}[\mathcal{L}_{A \cup B}(\theta + \Delta\theta_{\text{EFC}}(x_B)) - \mathcal{L}_{A \cup B}(\theta)] \leq -\eta \, \mathbb{E}_{x_B}[\|G_B^\perp(x_B)\|_{\tilde{F}_A^{-1}}^2] + O(\eta \, \tilde{\lambda}_{\min}^{-1}) + O(\eta^2). \tag{S120}$$

*Proof.* A second-order Taylor expansion gives

$$\mathcal{L}_{A \cup B}(\theta + \Delta\theta) - \mathcal{L}_{A \cup B}(\theta) = \nabla_\theta \mathcal{L}_{A \cup B}(\theta)^\top \Delta\theta + O(\|\Delta\theta\|^2). \tag{S121}$$

Apply the identity $\nabla_\theta \mathcal{L}_{A \cup B} = \nabla_\theta \mathcal{L}_A + \nabla_\theta \mathcal{L}_B$ and substitute $\Delta\theta = \Delta\theta_{\text{EFC}}(x_B)$. Use Lemma D.5 to write $\Delta\theta_{\text{EFC}}(x_B) = -\eta \tilde{F}_A^{-1} G_B^\perp(x_B) + \varepsilon(x_B)$.

For the Task B contribution,

$$\nabla_\theta \mathcal{L}_B(x_B)^\top \Delta\theta_{\text{EFC}}(x_B) = (G_B^\perp(x_B) + G_B^\parallel(x_B) + G_{A \leftarrow B}(x_B))^\top \left( -\eta \tilde{F}_A^{-1} G_B^\perp(x_B) + \varepsilon(x_B) \right) \tag{S122}$$

$$= -\eta \, (G_B^\perp(x_B))^\top \tilde{F}_A^{-1} G_B^\perp(x_B) + (G_B^\parallel(x_B) + G_{A \leftarrow B}(x_B))^\top \varepsilon(x_B) \tag{S123}$$

$$+ (G_B^\perp(x_B))^\top \varepsilon(x_B) - \eta \, (G_B^\parallel(x_B) + G_{A \leftarrow B}(x_B))^\top \tilde{F}_A^{-1} G_B^\perp(x_B). \tag{S124}$$

The first term equals $-\eta \|G_B^\perp(x_B)\|_{\tilde{F}_A^{-1}}^2$. The final term is bounded by $O(\eta \, \tilde{\lambda}_{\min}^{-1})$ since $G_B^\parallel(x_B) + G_{A \leftarrow B}(x_B) \in \mathcal{V}_A$ and Lemma D.4 bounds $\tilde{F}_A^{-1}$ on $\mathcal{V}_A$. The remaining terms are also bounded by $O(\eta \, \tilde{\lambda}_{\min}^{-1})$ using the bound on $\varepsilon(x_B)$ from Lemma D.5 and Cauchy-Schwarz.

For the Task A contribution, $\nabla_\theta \mathcal{L}_A(\theta) \in \mathcal{V}_A$ by definition of $\mathcal{V}_A$. The inner product $\nabla_\theta \mathcal{L}_A(\theta)^\top (-\eta \tilde{F}_A^{-1} G_B^\perp(x_B))$ is bounded by $O(\eta \, \tilde{\lambda}_{\min}^{-1})$ under the same restriction argument, and the contribution from $\varepsilon(x_B)$ is bounded by $O(\eta \, \tilde{\lambda}_{\min}^{-1})$.

Combine bounds and take expectation over $x_B$. The Taylor remainder contributes $O(\eta^2)$ under bounded second derivatives and bounded gradients. $\qquad\square$

## E. Forgetting Bounds

We derive bounds on forgetting—the increase in Task A's loss caused by learning Task B. For any parameter update $\Delta\theta = \theta - \theta_A^*$, the change in Task A's loss admits a quadratic approximation:

$$\mathcal{L}_A(\theta_A^* + \Delta\theta) - \mathcal{L}_A(\theta_A^*) = \frac{1}{2}\Delta\theta^\top F_A \Delta\theta + O(\|\Delta\theta\|^3), \tag{S125}$$

where $F_A$ is the Fisher information matrix at $\theta_A^*$. This approximation holds for small updates and follows from the Laplace approximation (Eq. S2). The forgetting bound for each method therefore depends on how its update $\Delta\theta$ interacts with Task A's curvature $F_A$.

### E.1. Update rules

Each method induces a different parameter update:

- BP performs gradient descent on Task B alone:

$$\Delta\theta_{\mathrm{BP}} = -\eta\nabla_\theta \mathcal{L}_B. \tag{S126}$$

- EWC minimizes the regularized objective $\mathcal{L}_B(\theta) + \frac{\beta}{2}(\theta - \theta_A^*)^\top D_A(\theta - \theta_A^*)$, where $D_A = \mathrm{diag}(F_A)$. The resulting deviation optimistically satisfies:

$$\Delta\theta_{\mathrm{EWC}} = -(\nabla^2 \mathcal{L}_B + \beta D_A)^{-1}\nabla_\theta \mathcal{L}_B. \tag{S127}$$

  When $\beta D_A \gg \nabla^2 \mathcal{L}_B$, this simplifies to $\Delta\theta_{\mathrm{EWC}} \approx -\frac{1}{\beta}D_A^{-1}\nabla_\theta \mathcal{L}_B$. Even with access to the full Fisher $F_A$ (the theoretically optimal quadratic regularizer), the update would be $\Delta\theta = -(\nabla^2 \mathcal{L}_B + \beta F_A)^{-1}\nabla_\theta \mathcal{L}_B$, which still couples Task B's Hessian with the regularizer.

- EFC learns at dynamical equilibrium, yielding the continual-natural gradient (Theorem 3.1):

$$\Delta\theta_{\mathrm{EFC}} = -\eta\tilde{F}_A^{-1}\nabla_\theta \mathcal{L}_B. \tag{S128}$$

  Here, Task A's curvature appears directly as a preconditioner, not additively combined with $\nabla^2 \mathcal{L}_B$. This structural difference explains why EFC can outperform even theoretically optimal Hessian regularization.

### E.2. Direct forgetting bounds

**Theorem E.1** (Direct Forgetting Bounds). *Let $\theta_A^*$ be the optimal parameters for Task A, $F_A$ the Fisher information matrix at $\theta_A^*$, $\tilde{F}_A$ the implicit approximation from Theorem 3.1, and $D_A = \mathrm{diag}(F_A)$. Let $\lambda_{\max}$ and $\tilde{\lambda}_{\max}$ be the maximum eigenvalues of $F_A$ and $\tilde{F}_A$, respectively. For learning rate $\eta < \min\{\frac{1}{\lambda_{max}}, \frac{1}{\|D_A\|_\infty}\}$, the forgetting bounds are:*

$$\mathcal{L}_A(\theta_A^* + \Delta\theta_{BP}) - \mathcal{L}_A(\theta_A^*) \leq \frac{\eta^2}{2}\|\nabla_\theta \mathcal{L}_B\|_2^2\,\lambda_{\max} \tag{S129}$$

$$\mathcal{L}_A(\theta_A^* + \Delta\theta_{EWC}) - \mathcal{L}_A(\theta_A^*) \leq \frac{\eta^2}{2}\|\nabla_\theta \mathcal{L}_B\|_2^2\,\mathrm{tr}(D_A^{-1}F_A D_A^{-1}) + O(\|\nabla^2 \mathcal{L}_B\|) + O(\|D_A - F_A\|) \tag{S130}$$

$$\mathcal{L}_A(\theta_A^* + \Delta\theta_{EFC}) - \mathcal{L}_A(\theta_A^*) \leq \frac{\eta^2}{2}\|\nabla_\theta \mathcal{L}_B\|_2^2\,\tilde{\lambda}_{\max}^{-1} + O(\|\tilde{F}_A - F_A\|), \tag{S131}$$

*where $\eta$ absorbs the regularization strength $\beta^{-1}$ for EWC.*

*Proof.* For BP with $\Delta\theta_{\mathrm{BP}} = -\eta\nabla_\theta \mathcal{L}_B$:

$$\frac{1}{2}\Delta\theta_{\mathrm{BP}}^\top F_A \Delta\theta_{\mathrm{BP}} = \frac{\eta^2}{2}(\nabla_\theta \mathcal{L}_B)^\top F_A \nabla_\theta \mathcal{L}_B \leq \frac{\eta^2}{2}\|\nabla_\theta \mathcal{L}_B\|_2^2\lambda_{\max}. \tag{S132}$$

For EWC, let $M = \nabla^2 \mathcal{L}_B + \beta D_A$. The exact update is $\Delta\theta_{\text{EWC}} = -M^{-1}\nabla_\theta\mathcal{L}_B$, so

$$\frac{1}{2}\Delta\theta_{\text{EWC}}^\top F_A \Delta\theta_{\text{EWC}} = \frac{1}{2}(\nabla_\theta\mathcal{L}_B)^\top M^{-1} F_A M^{-1} \nabla_\theta\mathcal{L}_B. \tag{S133}$$

When $\beta D_A$ dominates, $M^{-1} \approx (\beta D_A)^{-1} - (\beta D_A)^{-1}\nabla^2\mathcal{L}_B(\beta D_A)^{-1} + O(\|\nabla^2\mathcal{L}_B\|^2)$, yielding

$$\approx \frac{1}{2\beta^2}(\nabla_\theta\mathcal{L}_B)^\top D_A^{-1} F_A D_A^{-1}\nabla_\theta\mathcal{L}_B + O(\|\nabla^2\mathcal{L}_B\|) + O(\|D_A - F_A\|). \tag{S134}$$

The trace simplifies since $D_A = \text{diag}(F_A)$:

$$\text{tr}(D_A^{-1} F_A D_A^{-1}) = \sum_i \frac{[F_A]_{ii}}{[D_A]_{ii}^2} = \sum_i \frac{1}{[D_A]_{ii}} = \text{tr}(D_A^{-1}) \leq \frac{n}{\lambda_{\min}^D}, \tag{S135}$$

where $\lambda_{\min}^D = \min_i [D_A]_{ii}$ is the minimum eigenvalue of $D_A$. Therefore,

$$\leq \frac{\eta^2}{2}\|\nabla_\theta\mathcal{L}_B\|_2^2 \frac{n}{\lambda_{\min}^D} + O(\|\nabla^2\mathcal{L}_B\|) + O(\|D_A - F_A\|), \tag{S136}$$

with $\eta = 1/\beta$.

For EFC with $\Delta\theta_{\text{EFC}} = -\eta\tilde{F}_A^{-1}\nabla_\theta\mathcal{L}_B$:

$$\frac{1}{2}\Delta\theta_{\text{EFC}}^\top F_A \Delta\theta_{\text{EFC}} = \frac{\eta^2}{2}(\nabla_\theta\mathcal{L}_B)^\top \tilde{F}_A^{-1} F_A \tilde{F}_A^{-1}\nabla_\theta\mathcal{L}_B \tag{S137}$$

$$= \frac{\eta^2}{2}(\nabla_\theta\mathcal{L}_B)^\top \tilde{F}_A^{-1}\nabla_\theta\mathcal{L}_B + O(\|\tilde{F}_A - F_A\|) \tag{S138}$$

$$\leq \frac{\eta^2}{2}\|\nabla_\theta\mathcal{L}_B\|_2^2 \tilde{\lambda}_{\max}^{-1} + O(\|\tilde{F}_A - F_A\|). \tag{S139}$$

$$\square$$

### E.3. Pairwise forgetting comparisons

**Corollary E.2** (Pairwise Forgetting Comparisons). *Under the conditions of Theorem E.1,*

$$\mathcal{L}_A(\theta_A^* + \Delta\theta_{EWC}) - \mathcal{L}_A(\theta_A^* + \Delta\theta_{EFC}) \tag{S140}$$

$$\geq \frac{\eta^2}{2}\|\nabla_\theta\mathcal{L}_B\|_2^2 \Big[\text{tr}(D_A^{-1} F_A D_A^{-1}) - \tilde{\lambda}_{\max}^{-1}\Big] + O(\|\nabla^2\mathcal{L}_B\|), \tag{S141}$$

$$\mathcal{L}_A(\theta_A^* + \Delta\theta_{BP}) - \mathcal{L}_A(\theta_A^* + \Delta\theta_{EFC}) \tag{S142}$$

$$\geq \frac{\eta^2}{2}\|\nabla_\theta\mathcal{L}_B\|_2^2 (\lambda_{\max} - \tilde{\lambda}_{\max}^{-1}). \tag{S143}$$

*When $F_A$ has significant off-diagonal structure, $\text{tr}(D_A^{-1} F_A D_A^{-1}) > \tilde{\lambda}_{\max}^{-1}$, showing EFC strictly outperforms EWC. When $F_A \approx D_A$, the methods become equivalent.*

*Proof.* Using the bounds from Theorem E.1, the proof is straightforward by computing the differences:

$$\frac{\eta^2}{2}\|\nabla_\theta\mathcal{L}_B\|_2^2\text{tr}(D_A^{-1} F_A D_A^{-1}) - \frac{\eta^2}{2}\|\nabla_\theta\mathcal{L}_B\|_2^2\tilde{\lambda}_{\max}^{-1} = \frac{\eta^2}{2}\|\nabla_\theta\mathcal{L}_B\|_2^2 \left(\text{tr}(D_A^{-1} F_A D_A^{-1}) - \tilde{\lambda}_{\max}^{-1}\right) \tag{S144}$$

$$\frac{\eta^2}{2}\|\nabla_\theta\mathcal{L}_B\|_2^2\lambda_{\max} - \frac{\eta^2}{2}\|\nabla_\theta\mathcal{L}_B\|_2^2\tilde{\lambda}_{\max}^{-1} = \frac{\eta^2}{2}\|\nabla_\theta\mathcal{L}_B\|_2^2 \left(\lambda_{\max} - \tilde{\lambda}_{\max}^{-1}\right) \tag{S145}$$

When $F_A$ has significant off-diagonal terms, $\text{tr}(D_A^{-1} F_A D_A^{-1}) > \tilde{\lambda}_{\max}^{-1}$, which makes the difference positive, showing EFC outperforms EWC. When $F_A \approx D_A$, the difference approaches zero, showing the methods become equivalent. $\square$

### E.4. Geometric interpretation of forgetting

**Corollary E.3** (Geometric interpretation and best-case analysis). *Consider the best-case scenario for EFC where the $\nabla_\theta \mathcal{L}_B$ aligns with $v_1 = \frac{1}{\sqrt{n}}(1, \ldots, 1)^\top$, maximizing interaction effects. Then the ratio of parameter space volumes constrained by EFC versus EWC to achieve the same forgetting bound $\epsilon$ is:*

$$\frac{Vol(EFC)}{Vol(EWC)} = \left( \frac{\lambda_1}{tr(D_A)/n} \right)^{n/2} \tag{S146}$$

*with the best-case forgetting comparison:*

$$\mathcal{L}_A(\theta_A^* + \Delta\theta_{EWC}) - \mathcal{L}_A(\theta_A^* + \Delta\theta_{EFC}) \geq \frac{\eta^2}{2} \|\nabla_\theta \mathcal{L}_B\|_2^2 \left[ \frac{n}{tr(D_A)} - \frac{1}{\lambda_1} \right] + O(\eta^3) \tag{S147}$$

This volume ratio is the geometric difference between the methods. EFC's constraint region forms a hyperellipsoid that accounts for parameter interactions, while EWC's constraint forms a hypercube that treats parameters independently. Thereby, EFC's advantage over EWC grows with dimensionality $n$ - the volume ratio between ellipsoid and cube increases exponentially with dimension when strong parameter interactions are present. The ratio is maximized in this best case where all parameters interact equally through $v_1$.

*Proof.* The EFC update from Theorem C.1 defines an hyperellipsoid, while EWC defines a hypercube:

$$(\theta - \theta_A^*)^\top \tilde{F}_A (\theta - \theta_A^*) \leq \epsilon \tag{S148}$$

$$(\theta - \theta_A^*)^\top D_A (\theta - \theta_A^*) \leq \epsilon \tag{S149}$$

In our best-case scenario for EFC, $v_1$ has equal components, making the interaction effects maximal. The volume ratio is:

$$\frac{Vol(EFC)}{Vol(EWC)} = \sqrt{\frac{\det(D_A)}{\det(\tilde{F}_A)}} = \left( \frac{\lambda_1}{tr(D_A)/n} \right)^{n/2} \tag{S150}$$

In the best case, $\nabla_\theta \mathcal{L}_B$ aligns with $v_1 = \frac{1}{\sqrt{n}}(1, \ldots, 1)^\top$. Then:

$$(\nabla_\theta \mathcal{L}_B)^\top D_A^{-1} F_A D_A^{-1} \nabla_\theta \mathcal{L}_B = \|\nabla_\theta \mathcal{L}_B\|_2^2 \cdot v_1^\top D_A^{-1} F_A D_A^{-1} v_1 \tag{S151}$$

$$= \|\nabla_\theta \mathcal{L}_B\|_2^2 \cdot \frac{1}{n} \sum_{i,j} \frac{F_{ij}}{d_i d_j} \tag{S152}$$

$$= \|\nabla_\theta \mathcal{L}_B\|_2^2 \cdot n \frac{\sum_{i,j} F_{ij}}{(\sum_i d_i)^2} \tag{S153}$$

$$\leq \|\nabla_\theta \mathcal{L}_B\|_2^2 \cdot n \frac{tr(F_A)}{(tr(D_A))^2} \tag{S154}$$

$$= \|\nabla_\theta \mathcal{L}_B\|_2^2 \cdot \frac{n}{tr(D_A)} \tag{S155}$$

giving,

$$\mathcal{L}_A(\theta + \Delta\theta_{EWC}) - \mathcal{L}_A(\theta + \Delta\theta_{EFC}) \geq \frac{\eta^2}{2} \|\nabla_\theta \mathcal{L}_B\|_2^2 \left[ \frac{n}{tr(D_A)} - \tilde{\lambda}_{max}^{-1} \right] + O(\eta^3) \tag{S156}$$

The ratio $\frac{n}{tr(D_A)}$ grows linearly with dimension $n$ because when $\nabla_\theta \mathcal{L}_B$ aligns with the normalized all-ones vector $v_1$, EWC treats all directions independently while EFC accounts for their coupling through the Fisher matrix. This dimensional scaling directly relates to our geometric intuition: EWC approximates the Fisher ellipsoid with a hypercube, and the ratio between a hypercube's volume and its hyperellipsoid grows exponentially with dimension, explaining why EWC's performance gap relative to EFC becomes more severe in higher dimensions. $\square$

### E.5. Sample-specific preservation tightens expected bounds

The bounds above analyze a single update. When taking expectations over Task B samples $\mathcal{D}_B$, EFC gains an additional advantage from sample-specificity. The preservation signal $\gamma$ depends on current activations, which vary across samples. When a Task B sample strongly overlaps with Task A's important directions (high Fisher), $\gamma$ is large and preservation is strong; otherwise $\gamma$ is small, allowing free learning.

Formally, let $\gamma_{\text{eff}}$ denote the cumulative effect of the preservation signals propagated through the network (see Supp. B). The expected forgetting bound for EFC is tightened by a variance term:

$$\mathbb{E}_{\mathcal{D}_B}\left[\mathcal{L}_A(\theta_A^* + \Delta\theta_{\text{EFC}}) - \mathcal{L}_A(\theta_A^*)\right] \leq [\text{direct bound}] - c \cdot \text{Var}_{\mathcal{D}_B}(\gamma_{\text{eff}}) \tag{S157}$$

for some constant $c > 0$. Static regularizers like EWC provide no such variance reduction and additionally suffer from the persistent $O(\|\nabla^2 \mathcal{L}_B\|)$ interference term under expectation.

## F. Regimes of Task B curvature

During class-incremental training on Task $B$, the combined loss exhibits an initial decrease, followed by a stationary point, and subsequent increase. This behavior reflects the transition from a regime where Task $B$ learning dominates to one where Task $A$ interference prevails.

The time evolution of the combined loss along the training trajectory is given by:

$$\frac{d}{dt}\mathcal{L}^{CIL} = \nabla_\theta \mathcal{L}^{CIL} \cdot \dot{\theta} \tag{S158}$$

For gradient-descent-based methods where $\dot{\theta} \propto -\nabla_\theta \mathcal{L}_B$, this decomposes as:

$$\frac{d}{dt}\mathcal{L}^{CIL} = \underbrace{-\|\nabla_\theta \mathcal{L}_B\|^2}_{\text{Task } B \text{ improvement}} \underbrace{-\nabla_\theta \mathcal{L}_A \cdot \nabla_\theta \mathcal{L}_B}_{\text{Task } A \text{ interference}} \tag{S159}$$

The first term is strictly non-positive, representing progress on Task $B$. The second term captures the interference between tasks: when gradients are anti-aligned, as will inevitable happen, this term becomes positive, indicating harm to Task $A$.

The combined loss reaches a stationary point when $\frac{d}{dt}\mathcal{L}^{CIL} = 0$, yielding:

$$\|\nabla_\theta \mathcal{L}_B\|^2 = -\nabla_\theta \mathcal{L}_A \cdot \nabla_\theta \mathcal{L}_B \tag{S160}$$

Invoking the quadratic approximation $\nabla_\theta \mathcal{L}_A \approx F_A(\theta - \theta_A^*)$ near the Task $A$ optimum, this condition becomes:

$$\|\nabla_\theta \mathcal{L}_B\|^2 = -\nabla_\theta \mathcal{L}_B^\top F_A \Delta\theta \tag{S161}$$

where $\Delta\theta = \theta - \theta_A^*$. Rearranging:

$$\nabla_\theta \mathcal{L}_B^\top \left(\nabla_\theta \mathcal{L}_B + F_A \Delta\theta\right) = \nabla_\theta \mathcal{L}_B^\top \nabla_\theta \mathcal{L}^{CIL} = 0 \tag{S162}$$

Thus, the stationary point occurs precisely when the Task $B$ gradient becomes orthogonal to the combined gradient. This analysis can be characterized by three distinct training regimes.

First, early in Task $B$ training, the network operates on a relatively flat plateau of the Task $B$ loss landscape. The Task $B$ gradient $\nabla_\theta \mathcal{L}_B$ is large while the accumulated displacement $F_A \Delta\theta$ remains small. Consequently:

$$\|\nabla_\theta \mathcal{L}_B\|^2 \gg \left|\nabla_\theta \mathcal{L}_B^\top F_A \Delta\theta\right| \tag{S163}$$

and the combined loss decreases rapidly. Task $B$ learning proceeds with minimal interference.

Second, as training progresses, two competing effects reach equilibrium: the Task $B$ gradient diminishes as the network approaches a Task $B$ solution, while the interference term $F_A \Delta\theta$ accumulates. At the stationary point:

$$\|\nabla_\theta \mathcal{L}_B\|^2 = \left|\nabla_\theta \mathcal{L}_B^\top F_A \Delta\theta\right| \tag{S164}$$

The rate of Task $B$ improvement exactly balances the rate of Task $A$ degradation.

Third, beyond the stationary point, the network enters the high-curvature region of Task $B$'s loss landscape. The Task $B$ gradient continues to shrink while the interference term persists, yielding:

$$\|\nabla_\theta \mathcal{L}_B\|^2 < \left|\nabla_\theta \mathcal{L}_B^\top F_A \Delta\theta\right| \tag{S165}$$

The combined loss now increases, and continued training causes net harm to overall performance despite continued Task $B$ improvement.

The Task $B$ Fisher information $F_B = \mathbb{E}[\nabla_\theta \mathcal{L}_B \nabla_\theta \mathcal{L}_B^\top]$ governs the rate at which the learning signal decays:

$$\frac{d}{dt}\|\nabla_\theta \mathcal{L}_B\|^2 \approx -2\nabla_\theta \mathcal{L}_B^\top F_B \nabla_\theta \mathcal{L}_B \tag{S166}$$

When $\|F_B\|$ is small (flat landscape) learning continues. As $\|F_B\|$ grows (entering the "bowl" of the Task $B$ optimum), the gradient diminishes rapidly.

# G. Confusion matrices for class-incremental learning

To visualize how different continual learning methods perform across all classes, we generate normalized confusion matrices after training on the full sequence of tasks (Figure S1). For each method, we train the model sequentially on five tasks, each containing two digit classes from MNIST, and evaluate on the combined test set of all previously seen classes. The confusion matrices are normalized row-wise by true label to show the proportion of samples from each class that are correctly classified versus misclassified as other classes. In the class-incremental setting, oEWC exhibits severe catastrophic forgetting of earlier tasks, with nearly zero accuracy on classes from Tasks 0–3 while retaining high accuracy only on the most recently learned task. This manifests as strong off-diagonal concentrations where samples from earlier classes are systematically misclassified as classes from the final task. DER++ maintains more balanced accuracy across all classes through replay, though some degradation on intermediate tasks remains observable. EFC demonstrates intermediate behavior, with confusion distributed bidirectionally across the task sequence.

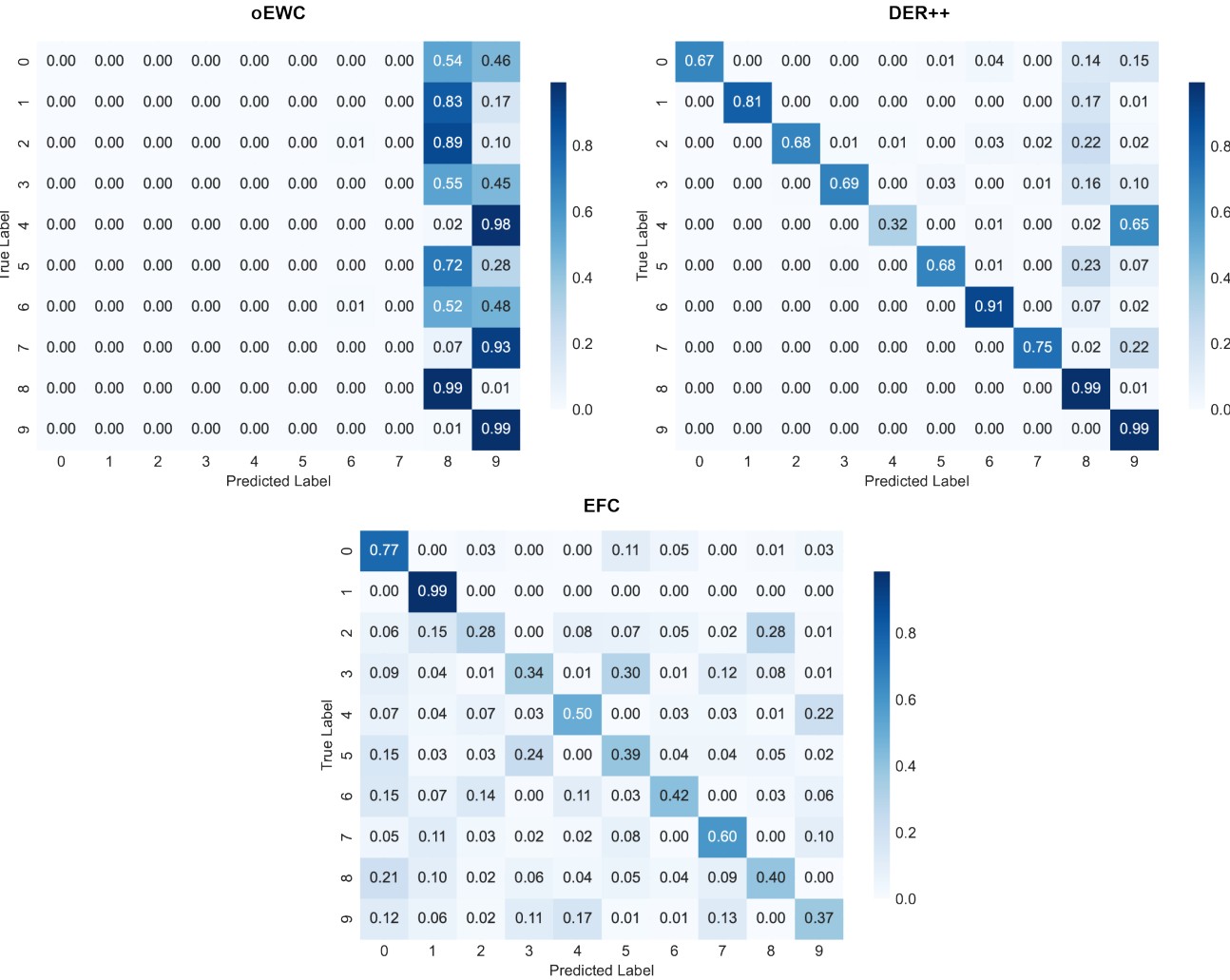

*Figure S1.* Confusion matrices on class-incremental Split-MNIST for online EWC (Schwarz et al., 2018) versus Dark Experience Replay++ (Buzzega et al., 2020) versus Equilibrium Fisher Control. For every matrix we plot the true label versus the predicted label. Values are measured in percentage of confusion.

# H. Hyperparameters and computational analysis

EFC inherits the dynamical systems machinery of the Deep Feedback Control (DFC) framework (Meulemans et al., 2021) and introduces one additional hyperparameter, the preservation strength $\beta$, that governs continual learning. We organize the hyperparameters into three groups: dynamics parameters inherited from DFC that govern how the network reaches equilibrium, the EFC-specific preservation strength, and standard training parameters. We then analyze computational overhead including per-step cost, memory, wall-clock time, and convergence behavior.

## H.1. Dynamics hyperparameters

The controlled dynamics (Eq. S26) require specifying how the network and controller evolve toward equilibrium. Following Meulemans et al. (2021), we parameterize the system with the quantities listed below. We adopt the implementation strategy of Meulemans et al. (2021) where the leak rate $\tilde{\alpha}$ is specified directly rather than derived from $\alpha$ and $k_p$, making this an independent hyperparameter.

**Proportional gain $k_p$.** The proportional control constant scales the instantaneous output error to produce the feedback signal. Larger $k_p$ drives the network more aggressively toward the output target, improving convergence speed but risking oscillatory instability when the network Jacobian $J$ is poorly conditioned. We use $k_p = 2.0$ across all experiments, consistent with the stable regime identified by Meulemans et al. (2021), which shows that performance is robust for $k_p \in [1.0, 3.0]$ and degrades outside this range due to slow convergence (small $k_p$) or oscillatory instability (large $k_p$).

**Controller leak rate $\tilde{\alpha}$.** The leak rate damps the integral component of the PI controller, preventing unbounded growth of the control signal and regularizing the dynamical inversion (Podlaski & Machens, 2020). In the EFC framework, $\tilde{\alpha}$ appears in the steady-state solution as $\mathbf{u}_\star = (J_{\text{eff}} + \tilde{\alpha}I)^{-1}(\delta_L^- - \gamma_{\text{eff}})$, acting as a Tikhonov regularizer on the effective Jacobian. Smaller $\tilde{\alpha}$ yields a more accurate dynamical inversion (closer to the pseudoinverse) but increases sensitivity to ill-conditioning. We use $\tilde{\alpha} = 0.0017$ for the empirical validation experiments and select $\tilde{\alpha}$ from $[10^{-4}, 10^{-1}]$ for the benchmark experiments. (Meulemans et al., 2021) shows that performance is stable across two orders of magnitude and degrades only for very large $\tilde{\alpha}$ where the damping suppresses learning.

**Time constant ratio $\tau_v/\tau_u$.** The ratio of the network time constant $\tau_v$ to the controller time constant $\tau_u$ determines the separation of timescales between the network and controller dynamics. The theoretical results of Meulemans et al. (2021) assume $\tau_v \ll \tau_u$ (fast network, slow controller), which ensures that the network settles before the controller updates. In practice, we use $\tau_v/\tau_u = 0.2$, which provides sufficient timescale separation for stable convergence while keeping the total number of iterations manageable.

**Simulation step size $\Delta t$ and maximum iterations $t_{\max}$.** The dynamics are simulated using forward Euler integration with step size $\Delta t$. Smaller $\Delta t$ improves numerical accuracy but increases the number of iterations required. We use $\Delta t = 0.02$ and $t_{\max} = 500$. In practice, convergence is typically reached well before $t_{\max}$: on Split-MNIST, while the initial learning steps can take hundreds of iterations, once the network is less random through training, the convergence is usually reached by tens or single-digit steps. The convergence threshold $\varepsilon = 10^{-4}$ determines when the dynamics have settled, measured as $\|\mathbf{u}^{(t+1)} - \mathbf{u}^{(t)}\| < \varepsilon$. Smaller epsilon gives higher precision, but convergence takes longer, this parameter can be relaxed to increase training speed.

**Output target step size $\lambda$.** Following DFC, the output target is defined as $\mathbf{r}_L^* = \mathbf{r}_L^- - \lambda \nabla_{\mathbf{r}_L} \mathcal{L}(\mathbf{r}_L^-)$. Smaller $\lambda$ makes the linearization underlying the theoretical results more accurate, while larger $\lambda$ provides stronger teaching signals. We use $\lambda = 0.02$ across experiments. Meulemans et al. (2021) showed that DFC performance is robust for $\lambda \in [10^{-3}, 10^{-1}]$, and we observe the same robustness in EFC.

## H.2. Preservation strength

The preservation strength $\beta > 0$ is the only hyperparameter introduced by EFC beyond the DFC framework. $\beta$ scales the preservation signal $\gamma_k = \beta \, \mathbf{r}_{i-1}^\top F_{A,k}^D (\theta_k - \theta_{A,k}^*)$, controlling how strongly prior-task curvature information modulates the network dynamics. The role of $\beta$ is analogous to the regularization strength in EWC, but its effect is qualitatively different: rather than adding a penalty to the loss, $\beta$ raises the cost of modulating specific neurons within the dynamics, forcing the learning signal to find alternative paths.

When $\beta = 0$, the preservation signal vanishes and EFC reduces to standard DFC without continual learning (Corollary A.4). As $\beta$ increases, the preservation signal increasingly constrains the dynamics, creating friction for the learning signal to modify neurons that were important for previous tasks. When $\beta$ is too large, the preservation signal dominates and new-task learning is suppressed entirely, as the control signal cannot overcome the preservation cost within the iteration budget.

We use $\beta = 0.05$ for the empirical validation experiments and select $\beta$ from $[10^{-3}, 10^{0}]$ for the benchmark experiments. The optimal operating point balances preservation and plasticity, and we find that $\beta \in [0.01, 0.1]$ consistently yields good performance across datasets.

We note that the preservation signal is normalized by the Fisher norm per neuron in our implementation, which reduces sensitivity to the absolute scale of $\beta$ across different network sizes and datasets. This normalization ensures that $\beta$ controls the relative strength of preservation versus learning rather than depending on the magnitude of the Fisher diagonal entries.

### H.3. Standard training hyperparameters

Beyond the dynamics and preservation parameters, EFC uses standard training hyperparameters: the learning rate for the Adam optimizer (Kingma & Ba, 2014), batch size, and number of training epochs per task. These are selected independently per method and per benchmark through hyperparameter search. For the benchmark experiments (Table 1), we train each task for 20 epochs with a batch size of 256 and select the learning rate from $[10^{-5}, 10^{-3}]$. All hyperparameter configurations are provided in our codebase.

### H.4. Computational overhead

**Per-step cost.** Each EFC training step consists of three phases (Algorithm 1). Phase 1 (feedforward pass) has the same cost as standard backpropagation: one forward pass through the network. Phase 2 (controlled dynamics) is the computational bottleneck: this requires up to $t_{\max}$ iterations, each consisting of a forward pass through all layers, a controller update, and computation of the preservation signal $\gamma$ for each layer. The preservation signal computation involves an element-wise multiplication of the presynaptic activities with the stored diagonal Fisher and parameter displacement, which is $O(|\theta|)$ per layer. Phase 3 (weight update) is a single outer product per layer, identical in cost to one gradient step. The total per-step cost is therefore approximately $t_{\mathrm{converge}}$ forward passes plus $t_{\mathrm{converge}}$ preservation signal computations, where $t_{\mathrm{converge}} \leq t_{\max}$ is the number of iterations until convergence.

In comparison, a standard backpropagation step requires one forward and one backward pass. EFC therefore incurs a factor of approximately $t_{\mathrm{converge}}$ overhead per training step. For our experiments, $t_{\mathrm{converge}}$ ranges from 80 to 200 depending on the difficulty of the sample and the stage of training. However, the realistic slowdown is much lower, as the GPU memory switching remains the bottleneck. The non-dynamical inversion mode (used for the benchmark experiments on Split-CIFAR10 and Split-Tiny-ImageNet) replaces the iterative dynamics with a single matrix solve of dimension $d_{\mathrm{output}} \times d_{\mathrm{output}}$, reducing the overhead to approximately 2–3× that of backpropagation at the cost of relying on the linearized steady-state approximation. The non-dynamical inversion scales cubic with respect to the output heads, so this is tractable for small tasks, yet detrimental for e.g. autoencoder settings, where the dynamical inversion has to be used.

**Memory cost.** EFC stores the diagonal of the Fisher information matrix for all previous tasks, requiring $O(|\theta|)$ additional memory, identical to EWC. No full curvature matrices or gradient subspaces are stored. During the dynamics, the additional memory consists of the controller state $\mathbf{u} \in \mathbb{R}^{d_L}$, the preservation signals $\gamma \in \mathbb{R}^N$, and the learning signals $\psi \in \mathbb{R}^N$, totaling $O(N + d_L)$ where $N$ is the total number of neurons and $d_L$ is the output dimension. For the feedback weights $Q_i$, we compute the Jacobian transpose $J_i^{\top}$ directly rather than storing separate feedback weight matrices, which adds no persistent memory overhead.

**Task boundary cost.** At each task boundary, EFC computes and stores the diagonal Fisher, requiring one pass through the training data with per-sample gradient accumulation. This is identical to the task-boundary cost of EWC and is negligible relative to the training cost.

**Wall-clock training time.** Table 2 reports wall-clock training times for a single task on each benchmark, measured on a single NVIDIA RTX 5090 GPU for Split-CIFAR10 and Split-Tiny-ImageNet. EFC with dynamical inversion is approximately 5–10× slower than backpropagation-based methods per training step due to the iterative dynamics. The non-dynamical inversion mode reduces this to approximately 2–5× overhead.

| Method | Split-MNIST | Split-CIFAR10 | Split-Tiny-ImageNet |
|---|---|---|---|
| BP / EWC / SI | ∼30s | ∼45s | ∼3min |
| EFC (dynamical) | ∼3min | ∼5min | ∼20min |

*Table 2.* Approximate wall-clock training time per task (20 epochs, batch size 256). BP-based methods include forward and backward pass only. EFC (dynamical) runs the full iterative dynamics to convergence.

**Scaling considerations.** The computational bottleneck of EFC is the iterative dynamics, which scales linearly with the number of neurons $N$ per iteration but requires multiple iterations to converge. For the frozen-encoder experiments (Split-CIFAR10, Split-Tiny-ImageNet), the dynamics operate only on the trainable MLP layers, making the overhead independent of the encoder size. For end-to-end training, the dynamics would need to propagate through the full network, and the per-iteration cost would scale with the depth and width of the network. As discussed in Section 5 of the main text, this computational cost is a fundamental limitation of dynamical systems approaches to learning and currently restricts EFC to settings where the trainable portion of the network is moderate in size.

