# OpenReview forum: "Continual Learning through Control Minimization"
_ICML.cc/2026/Conference — ICML 2026 regular_

### Official Review · Reviewer_DKoR · 2026-03-08

**Soundness:** 3
**Presentation:** 2
**Significance:** 3
**Originality:** 3
**Overall Recommendation:** 4
**Confidence:** 4

**Summary:**

This paper proposes EFC: it converts the regularization originally performed in parameter space for continual learning into a signal preservation mechanism in neural activity dynamics, allowing new task learning and old task retention to compete directly in an equilibrium state. Its core conclusion is that this approach can implicitly approximate the Fisher information of old tasks without explicitly storing the full curvature, thereby alleviating catastrophic forgetting. Furthermore, it outperforms traditional regularization methods under the replay-free setting.

**Compliance With Llm Reviewing Policy:**

Affirmed.

**Final Justification:**

The author's rebuttal has resolved most of my concerns, so I maintain my original positive evaluation.

**Key Questions For Authors:**

1. Does the current benchmark setup resemble continual learning on top of a fixed representation more than fully end-to-end continual learning? If so, what was the main motivation behind this design choice?
2. Could the authors provide a more complete analysis of computational overhead, such as the average number of equilibrium iterations per batch, wall-clock training time, memory cost, and whether the method still retains its advantages for larger models or end-to-end training?
3. The paper feels more theory-oriented overall, but it is still unclear to me how much this method can inform realistic large-scale continual learning systems. Could the authors elaborate more on its practical implications, as well as its potential and limitations in more realistic settings?

**Limitations:**

yes

**Strengths And Weaknesses:**

**Strengths:**
1. The paper has a clear methodological novelty: it shifts the stability objective in continual learning from parameter-space regularization to competition in activation dynamics, which is both original and conceptually interesting.
2. The theoretical part is strong and enjoyable to read. It not only explains the core intuition of the method, but also connects it to natural gradient descent and provides convergence and forgetting analyses.
3. Empirically, the method shows clear gains on class-IL over traditional regularization-based approaches, and notably narrows the gap to replay-based methods without using replay.


**Weaknesses:**
1. The benchmark is still somewhat toy-like, focusing mainly on image classification. In addition, some experiments are conducted with a frozen pretrained ResNet18 encoder rather than a fully end-to-end continual learning setting. I think the paper would be more convincing if the method were also validated on ViTs or other more modern end-to-end setups.
2. The experimental evaluation is still somewhat limited, and the baselines are mostly classic methods. The paper should ideally compare against two or three more recent SOTA methods, especially ones that are closer in spirit to the proposed approach.

---

> ### Author Rebuttal · Authors · 2026-03-30
>
> We thank the reviewer for the constructive feedback and the opportunity to clarify our design choices.
>
> Q1: The frozen-encoder design was intentional: we aimed to show that even when representations are sufficient for the task, parameter-based regularization still fails catastrophically in the class-incremental setting. In end-to-end training, one could argue that failures stem from poor representation learning rather than from the continual learning mechanism itself. By isolating the latter, we test our theoretical predictions directly. We wish to invite the reviewer to consider that our theory predicts no regularization method can resolve the class-incremental setting regardless of curvature quality (Section 3.2.1). This means newer state-of-the-art curvature estimation methods are equal in strength to the classical methods on this fundamental problem. To support this empirically, we additionally implemented Continual Learning With Quasi-Newton Methods (Vander Eeckt and Van Hamme, 2025), which represents the current state of the art in curvature-based continual learning and is close in spirit to our approach. As our theory predicts, this method remains completely unable to make progress on the class-incremental setting. We provide updated benchmark results here: https://ibb.co/FkRTk4Tf.
>
> Q2: We agree that a self-contained analysis of the implementation details, hyperparameters, and computational overhead should be present in the paper rather than deferred entirely to the DFC framework of Meulemans et al. (2021) which we rely on. Thus, we have added a new supplementary section covering all hyperparameters organized into three groups (dynamics parameters inherited from DFC, the EFC-specific preservation strength, and standard training parameters), their selection criteria and sensitivity, and a complete computational overhead analysis including per-step cost, memory, wall-clock training time, and scaling considerations. You can find this overview here: https://ibb.co/Ld6NTpQ7 (page 1), https://ibb.co/rfZQdjXg (page 2), https://ibb.co/Qsm8F9z (page 3).
>
> Q3: We agree that extending this line of research toward modern architectures such as ViTs is a natural ambition. However, we wish to gently push back on the framing that these benchmarks are toy-like. Current continual learning methods achieve around 10% accuracy on class-incremental Tiny-ImageNet, which is barely above chance for 200 classes. This suggests these benchmarks remain very much the frontier, and implementing any method on ViTs before solving these problems would unnecessarily overcomplicate the picture without addressing the underlying failure mode. Our theoretical contributions aim to steer the field toward dynamical learning frameworks and away from backpropagation-based approaches for avoiding catastrophic forgetting. At the current state of the field, this means modern architectures and large-scale problems are out of reach, and this is not specific to our framework but a fact of the field. The underlying control framework is architecture-agnostic and has been implemented on CNNs and RNNs by the original authors, so the extension is conceptually straightforward. The bottleneck is computational as the iterative dynamics scale with the depth of the trainable network. We discuss this limitation openly in Section 5 and provide a non-dynamical inversion mode that reduces overhead substantially.

---

> > ### Author Rebuttal · Reviewer_DKoR · 2026-04-02
> >
> > The rebuttal clarifies the rationale behind using a frozen encoder to isolate specific failure modes in class-incremental learning, and the results align with this theoretical framework. Nevertheless, I still have concerns about how this approach translates to end-to-end or larger-scale systems. Since the paper remains focused on traditional benchmarks and admits to being a proof-of-concept with high computational costs, its broader utility is not yet fully clear. I believe the authors have addressed my questions in part, but the final manuscript should more clearly emphasize these scope limitations and scalability trade-offs.

---

> > > ### Author Response · Authors · 2026-04-03
> > >
> > > We thank the reviewer for the follow-up and agree that the manuscript should be more explicit about scope limitations and scalability trade-offs. We have added a note at the end of Section 4.2.2., the revised text now reads:
> > >
> > > "We note that our evaluation trains moderate-sized MLPs either from scratch or on top of frozen encoders, and that scaling the controlled dynamics to fully end-to-end training on modern architectures remains an open challenge."
> > >
> > > and we have revised the Discussion as:
> > >
> > > "We view the current work as a proof-of-principle that continual learning can be reformulated as a control minimization problem and approached through a dynamical systems lens, but several limitations warrant discussion. First, dynamical systems approaches are substantially more computationally expensive than standard backpropagation, as each learning step requires running network dynamics to equilibrium through multiple iterations of forward and feedback passes. The per-iteration cost scales linearly with the number of trainable neurons, and convergence speed depends on the conditioning of the network Jacobian, which tends to worsen with depth, limiting applicability to problems where control-based methods remain tractable. Our current experiments train moderate-sized MLPs, either from scratch or on top of frozen encoders, and a gap remains between our evaluation and large-scale end-to-end continual learning on deep architectures. The broader utility of this work is therefore currently in establishing that control-based learning can address failure modes that no parameter-based regularization method can resolve, rather than in immediate practical deployment. Bridging this gap will require reducing the cost of running dynamics to equilibrium, which we consider the primary open challenge for control-based approaches to learning. Second, ..."
> > >
> > > We hope the reviewer finds this framing satisfactory and that it addresses the concerns raised. We agree with the reviewer that there should be no confusions about our framework's limits to scale. We also hope the reviewer agrees that continual learning remains unsolved even at the scale of our benchmarks, and that resolving these fundamental theoretical questions is a necessary step before scaling to larger architectures can be meaningful.

---

### Official Review · Reviewer_M1p8 · 2026-03-11

**Soundness:** 3
**Presentation:** 2
**Significance:** 4
**Originality:** 4
**Overall Recommendation:** 4
**Confidence:** 2

**Summary:**

This paper introduces Equilibrium Fisher Control to reformulate continual learning as a control problem within neural activity dynamics. By converting parameter-space regularization into activity-space preservation signals, the authors allow learning and preservation to compete dynamically. At equilibrium, weight updates implicitly capture the full prior-task curvature without requiring the explicit storage of large curvature matrices. The authors provide through theoretical analyses and experiments for the EFC.

**Compliance With Llm Reviewing Policy:**

Affirmed.

**Final Justification:**

The reviewer has resolve my concern. So I propose to weak accept the paper.

**Key Questions For Authors:**

1. Given the computational cost of reaching equilibrium, would it be possible to scale to larger models, like CNN or ViT?
2. Since the diagonal Fisher remains static, did you observe a performance drop as the number of tasks increases?

**Limitations:**

yes

**Strengths And Weaknesses:**

**Strengths**
1.  The paper provides a rigorous mathematical derivation of the Equilibrium Fisher Control framework. The authors establish formal forgetting bounds and solid proof for the theorems.
2. The empirical validation in Section 4.1 as well as Section G provides deep insight into how the training algorithm fulfill the theoretical insights.


**Weaknesses**
1. The paper often introduces complex variables and operators within equations (e.g., in Sections 2.1 and 2.2, Appendix A, B) without a centralized or preliminary notation guide.
2. While the theoretical framework is detailed, there is no high-level pseudocode or step-by-step practical algorithm illustrating the implementation for a standard training loop.

---

> ### Author Rebuttal · Authors · 2026-03-30
>
> We thank the reviewer for the detailed and constructive feedback, which has led to several concrete improvements in the manuscript.
>
> We have added a notation guide as the first page of the supplementary material, collecting all symbols in a table organized by category: network and parameters, neural activities and dynamics, control and preservation signals, curvature and Fisher information, gradient decomposition, and Jacobians and steady-state quantities. You can find it here: https://ibb.co/FqdF74VP. We hope this is sufficient and welcome further suggestions. We have also improved the expository text in Sections 2.1 and 2.2, clarifying that f(phi, theta) denotes the concatenated feedforward computation across all layers, that the exponential e^(psi + gamma) is element-wise rather than a matrix exponential, and making the connection between the dynamics equation and the equilibrium constraint explicit. We want to thank the reviewer for this suggestion, because as authors who have spent many months inside this formalism, we had become blind to how the notation reads on a first pass. Additionally, we have added a high-level pseudocode algorithm (Algorithm 1) in Supplementary Section A.4, which you can find here: https://ibb.co/3g28Rqk. The pseudocode illustrates the full EFC training loop, making the three-phase structure of each learning step explicit. We hope this provides an accessible entry point into the implementation without requiring the reader to consult the codebase.
>
> Q1: This question was raised by multiple reviewers and we address it in detail in our responses to Reviewers DKoR (Q3) and 8KJH (Q3). In brief, the underlying control framework is architecture-agnostic and has been implemented on CNNs and RNNs by the original authors. The computational bottleneck is the iterative dynamics, whose cost scales with the depth of the trainable portion of the network. We have added a new supplementary section with a complete computational overhead analysis. You can find a draft of this overview here: https://ibb.co/Ld6NTpQ7 (page 1) https://ibb.co/rfZQdjXg (page 2) https://ibb.co/Qsm8F9z (page 3).
>
> Q2: This is an important concern, and closely related to the progressive curvature misalignment experiment we added in response to Reviewer 8KJH (Q1). We refer the reviewer to that response and to the expanded Figure 1 (https://ibb.co/nM5BNW2K), where we quantify how stored curvature approximations drift from the true local geometry during training. The stored full Fisher loses roughly 50% of its structural alignment within ten epochs, and the diagonal Fisher degrades further still. As tasks accumulate, this misalignment compounds. We acknowledge this as a limitation: EFC mitigates this largely as seen in Fig. 1d (see Reviewer 8KJH (Q1)), as the off-diagonal structure is recovered dynamically while the diagonal remains static. A fully dynamical approach that derives all curvature information from the network dynamics without storing any curvature explicitly is a natural direction for future work that we discuss in Section 5.

---

> > ### Author Rebuttal · Reviewer_M1p8 · 2026-04-04
> >
> > The response is clear and extensive. It has resolved my concern.

---

### Official Review · Reviewer_8KJH · 2026-03-11

**Soundness:** 3
**Presentation:** 3
**Significance:** 3
**Originality:** 3
**Overall Recommendation:** 5
**Confidence:** 3

**Summary:**

This paper addresses the problem of catastrophic forgetting in sequential learning. To minimize the interference that learning a new task has on the representation of preceding tasks, the authors formulate continual learning as a control problem in which learning and preservation signals protecting prior-task representations compete within neural activity dynamics. The learning objective follows the least-control principle, minimizing the control effort needed to integrate new tasks while preserving prior tasks. Thus, the proposed framework (Equilibrium Fisher Control) couples learning and preservation signals during the formation of learning signals to ensure that learning of new tasks is coupled to the preservation of prior tasks.
When this framework is at equilibrium, learning occurs within the geometry of previously learned tasks because neural activities produce weight updates that implicitly encode the full prior task-curvature. The authors name this the “continual-natural learning gradient’.

The Equilibrium Fisher Control (EFC) framework converts parameter-space regularizers into activity-dependent preservation signals that modulate neural activities, ensuring the learning signal competes with preservation signals before the parameters update. By projecting the regularizer gradient onto the current pre-synaptic activity, a neuron-specific preservation signal is construed that activates only when the current input engages neurons that were involved in a prior task. The network dynamics are formulated to include an exponential multiplicative which scales modulation with current neural activity and couples learning and preservation. The learning objective is to minimize the learning signal required to reach a loss-minimizing equilibrium while competing with the preservation signal. In this set-up, a direction that modifies a prior representation is expensive, while directions orthogonal to prior learning are cheap.

The paper formalizes several learning-theory predictions for the EFC framework, starting with the continual-natural gradient property and the class-incremental convergence that enables task discrimination. Next, the paper posits that the proposed framework can counteract the interference term better than a parameter-based regularizer, because the latter computes only a single vector and therefore cannot mitigate per-sample variations in interference while in the proposed framework the interference term depends on the activations induced by the specific input sample and thus counteracts the sample-specific interference term. Finally, the paper shows that the proposed EFC framework has tighter forgetting bounds than standard backpropagation or elastic weight conditioning (EWC), because EFC captures off-diagonal curvature explicitly (avoiding dimensional scaling) and because EFC utilizes the aforementioned per-sample preservation.

The theoretical predictions are validated through a small, controlled experimental setup in which computation of the full Fisher (for comparison) is tractable. For this, a two-layer MLP and the two-task Split-MNIST. The EFC framework is compared against standard backpropagation, EWC and an EWC variant in which the diagonal Fisher is replaced with the full Fisher to represent optimal regularization (FISH). The results confirm that the continual-natural gradient in EFC approximates the full Fisher, that EFC achieves stronger class-incremental convergence, and that dynamic storage outperforms static storage (by avoiding the progressive misalignment issue).

In a second experiment, the performance of EFC is evaluated against other parameter-based regularization methods as well as an experience replay method across task-incremental and class-incremental settings. The results show that the EFC framework outperforms these parameter-regularization methods in task-incremental settings on the Split-CIFAR10 and Split-Tiny-ImageNet, but not on the SplitMNIST. In class-incremental settings, EFC shows a large performance jump in comparison to the parameter-regularization methods. Experience replay still performs significantly better, but EFC begins to approach the performance of replay approaches for the first time.

The paper is concluded by presenting a biological motivation for the proposed framework focused on the properties of pyramidal neurons in Layer 5 of cortex,

**Compliance With Llm Reviewing Policy:**

Affirmed.

**Final Justification:**

This is is a solid paper and the rebuttal addressed reviewer points sufficiently. I stand by my assessment to accept.

**Key Questions For Authors:**

•	The paper hypothesizes that the reason for the lower accuracy for the full Fisher in comparison with the EFC framework despite similar loss progression is a consequence of the curvature information becoming increasingly misaligned when explicitly stored as in the full Fisher approach. As this progressive misalignment of the curvature (and EFC’s solution through dynamic estimation) is a central claim of the paper, a quantification of this progressive misalignment is warranted to provide an empirical proof of this point. Note also that the paper presents this point as a statement rather than a hypothesis in the header of section 4.1.3.
•	How are the optimal parameters (preservation strength, controller leak rate, etcetera) selected and what is their impact on learning performance?
•	The proposed framework was evaluated on fully connected architectures only (that is, the ResNet-18 encoder was frozen, only the linear head was trained in this manner). Can the authors describe their expectations for this method for other types of architectures such as CNNs (sparse connectivity, weight sharing), as well as RNNs (temporal structure, implicit memory)?

**Limitations:**

Yes

**Strengths And Weaknesses:**

Soundness: The paper appears technically sound, providing a rigorous formulation of the theorems and methods. The combined theoretical motivation and empirical validation strengthen the foundation for the work.
Presentation: The manuscript has a clear writing style and narrative. The embedding in the context of existing work in the Introduction is appropriate and the distinction to prior research is presented in a clear manner. Information about the selection and impact of the hyperparameters of the dynamics is lacking though. For example, the selection and impact of preservation strength and controller leak rate is unclear. As the selection of these hyperparameters contributes to the complexity of the method (one of the key weaknesses of this approach), elucidating their selection and impact is not only relevant for reproducing the present work but also for setting up new experimentation.
Significance: The paper addresses the challenge of catastrophic forgetting in continual learning, a relevant and challenging topic in the field of machine hearing. The outcomes provide valuable insight into the effects of coupling learning and preservation signals during the formation of learning signals within neural activity dynamics as well as the benefit of per-sample interference estimation, thereby making an important contribution to the field. The scope of the impact is specialized rather than broad as the computational challenges of this method prohibit broad application in large-scale networks (as acknowledged also by the authors in the Discussion). Nevertheless, the paper constitutes a compelling proof-of-concept and identifies crucial new research directions.
Originality: The proposed reformulation of continual learning as a continual optimization problem presents a novel perspective on the challenge of sequential learning. A well-articulated theoretical foundation is provided and subsequently validated through experimentation, resulting in new insights and improved understanding of critical factors in continual learning.

---

> ### Author Rebuttal · Authors · 2026-03-30
>
> We appreciate the reviewer's careful reading, thorough feedback, and the positive assessment of our work.
>
> Q1: The reason we initially presented the progressive misalignment of curvature as a qualitative observation rather than a quantitative result is that EFC's curvature approximation is never materialized as an explicit matrix. It emerges implicitly from the interaction of preservation and learning signals within the network dynamics, and retrieving an explicit form from these signals is exceptionally difficult. However, after reading your review, we decided that you were correct, our paper makes a statement that ties into the central claim of our work, and we should provide empirical proof. We have revisited the problem and designed an experiment that quantifies both the progressive misalignment of stored curvature and EFC's robustness to it, without requiring access to the implicit approximation in explicit form. We have softened the header of Section 4.1.3 accordingly. The new results are included in the expanded Figure 1, which you can find here: https://ibb.co/nM5BNW2K. Plot 1c quantifies curvature drift directly. At every training step, we compute the true Fisher at the current parameters using Task A data over the full 10-class output head, then compare it to three static matrices: the identity, the stored diagonal Fisher, and the stored full Fisher, all computed once at the Task A optimum. The left panel shows the normalized Frobenius inner product measuring structural alignment; the stored full Fisher loses roughly 15% alignment within one epoch and 50% within ten epochs. The right panel shows the normalized Frobenius distance, with all static approximations converging toward a relative distance of 1.0. Plot 1d measures the consequences of this drift. Since we cannot observe the implicit approximation directly, we instead ask: does each method avoid Task A's sensitive directions? We compute the Rayleigh quotient of each method's cumulative parameter displacement with respect to the true current Fisher at that method's own parameters. Each method follows its own trajectory with independently computed true Fishers, ensuring a fair comparison. The left panel shows EFC maintains dramatically lower effective curvature throughout training. The right panel normalizes by the largest eigenvalue, bounding the quantity in [0,1]. EFC operates near zero while BP, EWC, and FISH cluster three orders of magnitude higher.
>
> Q2: We have added a new supplementary section (Supplementary H) with a complete analysis of all hyperparameters, organized into dynamics parameters inherited from the DFC framework, the EFC-specific preservation strength, and standard training parameters. We discuss selection criteria, sensitivity, and practical guidance for each. We additionally provide a computational overhead analysis covering per-step cost, memory, wall-clock training time, and scaling considerations. You can find a draft of this overview here: https://ibb.co/Ld6NTpQ7 (page 1) https://ibb.co/rfZQdjXg (page 2) https://ibb.co/Qsm8F9z (page 3). We refer the reviewer to our response to Reviewer DKoR for further discussion.
>
> Q3: We wish to clarify that EFC does not train only a linear head. For Split-CIFAR10 and Split-Tiny-ImageNet, a three-layer MLP with a hidden layer plus linear classifier is trained on top of the frozen encoder, and for Split-MNIST the full three-layer network is trained from scratch. The underlying control framework we adapt was designed for general network structures, and the original authors have implemented it on both CNNs and RNNs. The requirement is that individual neurons can be modulated and that teaching signals can be distributed to each neuron's weights. For CNNs, this is achieved by unrolling and distributing the convolution to the shared weights. The computational cost of running dynamics to equilibrium is the primary bottleneck for scaling, as we discuss in our limitations and the discussion above, but the framework itself is architecture-agnostic.

---

> > ### Author Rebuttal · Reviewer_8KJH · 2026-04-02
> >
> > Based on the response I stand by my original evaluation.

---

### Official Review · Reviewer_p3nV · 2026-03-12

**Soundness:** 4
**Presentation:** 4
**Significance:** 3
**Originality:** 4
**Overall Recommendation:** 5
**Confidence:** 3

**Summary:**

The paper addresses the catastrophic forgetting issue in neural networks. Inspired by neuroscience,  the paper reframes continual learning as a control problem in which learning and preservation signals (signal that protect prior-task representations) compete within neural activity dynamics. The training is done by minimising the control effort to learn new tasks while competing with the preservation signals. The preservation signal is constructed from the diagonal of the Fisher matrix, which measures per-parameter sensitivity on the previous task but implicitly encodes the full Fisher ( at equilibrium, neural activities give weight updates that implicitly encode the full prior-task curvature). The authors show that the method recovers the true prior-task curvature and outperforms existing methods on standard benchmarks for task-incremental and class-incremental learning.

**Compliance With Llm Reviewing Policy:**

Affirmed.

**Final Justification:**

I maintain my positive evaluation. This is a technically strong paper with high impact.  I did not increase my score as I believe there are still some limitations to the work (eg. Application to large models) , which does not make it ‘a technically flawless paper with exceptional impact’.  I recommend this paper for Acceptance.

**Key Questions For Authors:**

I have a few questions for the authors.

1. Could you explain why your method shows greater variance in some datasets? Where could this come from? Are all the improvements in the benchmark statistically significant?
2. To the best of your knowledge, in the brain, do you think that a combination of replay and control is used? What are the pros and cons of each of these two methods in artificial networks?
3. Do you have any intuition/idea on how to extend these ideas to large real networks?
4. The paper by Jarvis et al. (2025) shows that EWC does not always work due to the representation build by the network. Do you expect your method to have the same issue? More broadly what could be the strong failure cases of this method?

I look forward to your answer.

**Limitations:**

The paper discusses clearly the limitations and further work.

**Strengths And Weaknesses:**

I would like to first thank the author for their effort and for this interesting work.

**Soundness:**

**Strengths:** The paper is technically sound, and the claims are well supported by theoretical and experimental analysis.

**Presentation**

**Strengths:** The paper is clearly written, well structured, and the figures aid understanding.

**Weaknesses:** The author could add an introductory figure to visually depict the method to improve readability and understanding.

**Significance**

**Strengths:**  The paper is interesting and highly relevant to the ICML community. In my view, the main contribution is bridging the gap in incremental learning performance between parameter-based learning and the replay method using a novel approach.

**Weaknesses:**

As the authors state, this method is not applicable to large models (used in practice today), which limits its significance.

**Originality**

**Strengths:** The paper is part of a strong line of biologically inspired methods that bridges the fields of neuroscience and machine learning. The paper makes a clear prediction for neuroscientists to test. This model is relevant to both machine learning and neuroscience.

---

> ### Author Rebuttal · Authors · 2026-03-30
>
> We thank the reviewer for the thoughtful questions and the positive assessment of our contribution.
>
> Q1: The higher variance of EFC in class-incremental settings reflects the fact that EFC finds non-trivial solutions where baselines converge to near-random accuracy with minimal variance. The root cause is anchor quality. The initial stored Fisher determines the quality of the preservation signal for all subsequent tasks, and small differences in first-task performance compound across the task sequence. We conducted two-sample Welch t-tests (n=5 per method) comparing EFC against the strongest baseline per setting. In Class-IL, improvements are significant across all benchmarks: Split-MNIST 51.4 vs 20.2 (p<0.001), Split-CIFAR10 50.2 vs 21.2 (p<0.001), Split-Tiny-ImageNet 8.8 vs 4.8 (p<0.001). These gaps far exceed the combined uncertainty and are consistent with our theoretical prediction that parameter-based regularization cannot resolve the class-incremental setting regardless of curvature quality (Section 3.2). In Task-IL, the picture is more nuanced. On Split-Tiny-ImageNet, EFC improves significantly (37.2 vs 35.5, p<0.01). On Split-CIFAR10 and Split-MNIST, differences are modest and not significant, which is expected, task identifiers eliminate the interference term that EFC is designed to filter. We note that formal significance testing is not standard practice in continual learning benchmarking, and to our knowledge none of the baselines report p-values in their original publications.
>
> Q2: To the best of our knowledge, replays (or reactivations, as they are called in cortex) implement a control-like learning algorithm similar to the one we propose. Aceituno et al. 2025 tested this explicitly on data from Nguyen et al. 2024. The target learning framework we built upon was designed to work within these same cortical dynamics and learning rules (Meulemans et al. 2021, Aceituno et al. 2025). Regarding the pros and cons, one motivation behind our work was the difficulty of defining replay in a biologically faithful way. In artificial continual learning, replay means storing and re-presenting raw inputs, but in the brain, reactivations are internally generated patterns driven by hippocampal-cortical interactions, often during rest or sleep (Rothschild et al. 2017). A future direction we mention in the paper is extending our framework so that the dynamics themselves generate implicit functional replays without an explicit buffer. Regarding pros and cons in artificial networks, replay is simple, scales well, and performs strongly, but requires storing raw data (raising privacy and memory concerns, a whole subfield) and introduces buffer selection biases. Control-based methods require no stored data and operate within the geometry of learned representations, but are computationally expensive and introduce additional hyperparameters.
>
> Q3: Due to rebuttal length constraints and because this question was raised by multiple reviewers, we direct the reviewer to our responses to Reviewers DKoR (Q3) and 8KJH (Q3).
>
> Q4: Thank you for pointing us to this work. If we understand the reviewer's question correctly, the result of Jarvis et al. relevant to our work is their finding in Section 4.2.2, when a network fails to specialise (i.e., multiple neurons learn redundant representations of the same feature), EWC cannot distinguish between these redundant weights and either over-regularises (preventing all learning) or under-regularises (preventing all preservation), with no useful operating point in between. Since EFC constructs its preservation signal from the same diagonal Fisher, it inherits this degeneracy at the level of stored curvature. However, EFC's dynamics provide a partial remedy. The preservation signal is activity-dependent and activates only when the current input engages synapses relevant to previous tasks. If redundant neurons receive different presynaptic inputs on a given Task B sample, the preservation signal differentiates between them on a per-sample basis, breaking the symmetry that Jarvis et al. identify as the core failure mode. We therefore expect EFC's failure cases to arise where the preservation signal cannot distinguish between neurons, highly redundant representations combined with highly overlapping input distributions across tasks.

---

> > ### Author Rebuttal · Reviewer_p3nV · 2026-04-02
> >
> > Thank you to the authors for their thoughtful response; I will maintain my positive evaluation.

---

### Decision · Program_Chairs · 2026-04-30

**Decision:**

Accept (regular)

**Comment:**

This work proposes a framework Equilibrium Fisher Control (EFC), which reframes replay-free continual learning (CL) as a control problem in neural activity dynamics. Notably, this is in contrast to much of the prior work that addresses CL via parameter-space regularization. The main claim in this work is that this dynamical formulation allows updates to implicitly capture full prior-task curvature at equilibrium, while requiring only diagonal Fisher storage. All reviewers found the paper's idea to be original, theoretically strong, well-motivated, useful for the analysis of class-incremental learning and replay-free continual learning.

The reviewers were unanimously positive about the paper and maintained their positive recommendations after rebuttal. The author rebuttal also addressed most of the reviewers' concerns adequately, especially by clarifying the curvature-misalignment claim, adding discussion of hyperparameters and computational overhead. The presentation also improved with notation guidance and pseudocode. Reviewer DKoR did expressed some concern about practical relevance and scalability, but maintained their positive assessment.

The main weakness is that the empirical evaluation remains somewhat limited in scope:  moderate-scale experiments, sometimes with frozen encoders, and the method being too computationally heavy to demonstrate practical end-to-end, large-scale CL. While the paper should be more transparent about these limitations, it shouldn't be disqualifying, and I recommend the paper for acceptance. Nevertheless, for the camera-ready version, the authors are encouraged to clearly emphasize the current scalability limitations and frame the work as a proof-of-principle and theoretical advance rather than an immediately practical large-scale solution.